# Tailoring: encoding inductive biases by optimizing unsupervised objectives at prediction time

**Ferran Alet, Maria Bauza, Kenji Kawaguchi,**
**Nurullah Giray Kuru, Tomás Lozano-Pérez, Leslie Pack Kaelbling**
MIT
{alet,bauza,kawaguch,ngkuru,tlp,lpk}@mit.edu

## Abstract

From CNNs to attention mechanisms, encoding inductive biases into neural networks has been a fruitful source of improvement in machine learning. Adding auxiliary losses to the main objective function is a general way of encoding biases that can help networks learn better representations. However, since auxiliary losses are minimized only on training data, they suffer from the same generalization gap as regular task losses. Moreover, by adding a term to the loss function, the model optimizes a different objective than the one we care about. In this work we address both problems: first, we take inspiration from transductive learning and note that after receiving an input but before making a prediction, we can fine-tune our networks on any unsupervised loss. We call this process *tailoring*, because we customize the model to each input to ensure our prediction satisfies the inductive bias. Second, we formulate *meta-tailoring*, a nested optimization similar to that in meta-learning, and train our models to perform well on the task objective after adapting them using an unsupervised loss. The advantages of tailoring and meta-tailoring are discussed theoretically and demonstrated empirically on a diverse set of examples.

## 1 Introduction

The key to successful generalization in machine learning is the encoding of useful inductive biases. A variety of mechanisms, from parameter tying to data augmentation, have proven useful to improve the performance of models. Among these, auxiliary losses can encode a wide variety of biases, constraints, and objectives; helping networks learn better representations and generalize more broadly. Auxiliary losses add an extra term to the task loss that is minimized over the training data.

However, they have two major problems:

1. Auxiliary losses are only minimized at training time, but not for the query points. This leads to a generalization gap between training and testing, in addition to that of the task loss.

2. By minimizing the sum of the task loss plus the auxiliary loss, we are optimizing a different objective than the one we care about (only the task loss).

In this work we propose a solution to each problem:

1. We use ideas from *transductive learning* to minimize unsupervised auxiliary losses at each query, thus eliminating their generalization gap. Because these losses are unsupervised, we can optimize them at any time inside the prediction function. We call this process *tailoring*, since we customize the model to each query.

2. We use ideas from *meta-learning* to learn a model that performs well on the task loss after being tailored with the unsupervised auxiliary loss; i.e. *meta-tailoring*. This effectively trains the model to leverage the unsupervised tailoring loss in order to minimize the task loss.

35th Conference on Neural Information Processing Systems (NeurIPS 2021).

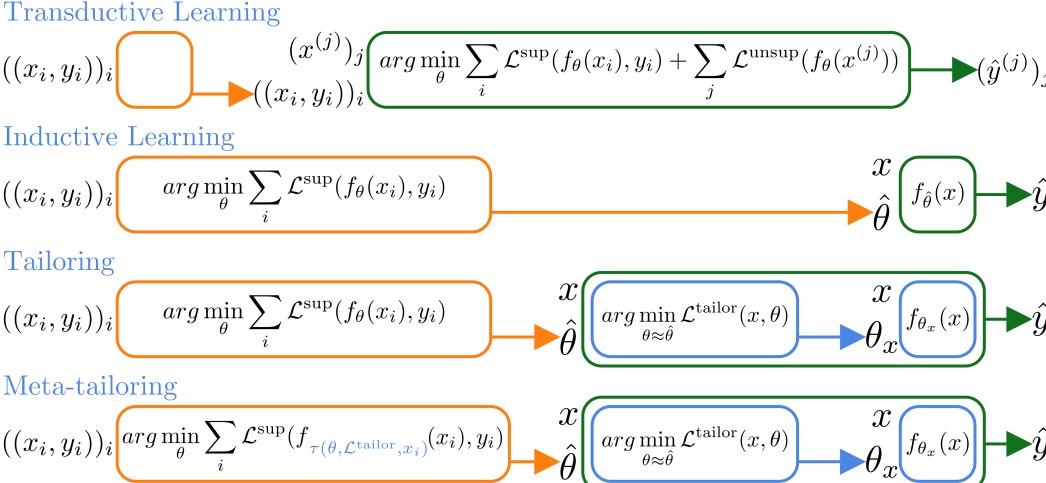

Figure 1: Comparison of several learning settings with *offline* computation in the orange boxes and *online* computation in the green boxes, with tailoring in blue. For meta-tailoring training, $\tau(\theta, \mathcal{L}^{\text{tailor}}, x) = \text{argmin}_{\theta' \approx \theta} \mathcal{L}^{\text{tailor}}(x, \theta')$ represents the tailoring process resulting in $\theta_x$.

**Illustrative example**   Imagine you want to use a neural network to predict the motion of a planetary system: given the positions and velocities of each planet, the network predicts their future positions and velocities. Additionally, we could encode energy and momentum conservation by adding an auxiliary loss encouraging the neural network to conserve energy and momentum for the training examples. However, this does not guarantee that the network will conserve them for test queries. Alternatively, we can exploit that evaluating these conservations requires comparing only the input with the prediction without needing access to the true target. Therefore, we can enforce these conservations by optimizing an unsupervised objective within the prediction function. In doing so, we *tailor* the model to each individual query to ensure it satisfies energy and momentum conservation. Taking into account this prediction-time adaptation during training leads to a two-layer optimization, where we train to make accurate predictions after encouraging the physical conservations.

**Tailoring a predictor**   Traditionally, supervised learning is approached within the inductive learning framework, shown in the second row of Figure 1. There, an algorithm consumes a training dataset of input-output pairs, $((x_i, y_i))_{i=1}^n$, and produces a set of parameters $\hat{\theta}$ by minimizing a supervised loss $\sum_{i=1}^n \mathcal{L}^{\text{sup}}(f_\theta(x_i), y_i)$ and, optionally, an unsupervised auxiliary loss $\sum_{i=1}^n \mathcal{L}^{\text{unsup}}(\theta, x_i)$. These parameters specify a hypothesis $f_{\hat{\theta}}(\cdot)$ that, given a new input $x$, generates an output $\hat{y} = f_{\hat{\theta}}(x)$. This problem setting misses a substantial opportunity: before the learning algorithm sees the query point $x$, it has distilled the data down to the parameters $\hat{\theta}$, which are frozen during inference, and so it cannot use new information about the *particular* $x$ that it will be asked to make a prediction for.

Vapnik recognized an opportunity to make more accurate predictions when the query point is known, in a framework that is now known as *transductive learning* [50, 11], illustrated in the top row of Figure 1. In transductive learning, a single algorithm consumes both labeled data, $((x_i, y_i))_{i=1}^n$, and a set of input queries for which predictions are desired, $(x^{(j)})_j$, and produces predictions $(\hat{y}^{(j)})_j$ for each query. In general, however, we do not know queries *a priori*, and instead, we want an inductive function that makes predictions online, as queries arrive. To obtain such an online prediction function from a transductive system, we would need to take the training data and the single unlabeled query and encapsulate the entire transductive learning procedure inside the prediction function itself. This strategy would achieve our objective of taking $x$ into account at prediction time but would be computationally much too slow [12].

This approach for combining induction and transduction would reuse the same training data and objective for each prediction, only changing the single unlabeled query. Consequently, it would perform extremely similar computations for each prediction. Therefore, we propose to effectively reuse the shared computations and find a "meta-hypothesis" that can then be efficiently adapted to each query. As shown in the third row of Figure 1, we propose to first run regular supervised learning to obtain parameters $\hat{\theta}$. Then, given a query input $x$, we fine-tune $\hat{\theta}$ on an unsupervised loss $\mathcal{L}^{\text{tailor}}$ to obtain cus-

**Algorithm 1 MAMmoTh: M**odel-**A**gnostic **M**eta-**T**ailoring

---

**Subroutine** *Training(f, $\mathcal{L}^{\text{sup}}$, $\lambda_{sup}$, $\mathcal{L}^{\text{tailor}}$, $\lambda_{tailor}$, $\mathcal{D}_{train}$, b)*
  randomly initialize $\theta$
  **while** *not done* **do**
   Sample batch of samples $(x_i, y_i) \sim \mathcal{D}_{train}$
   **forall** $(x_i, y_i)$ **do**
    $\theta_{x_i} = \theta - \lambda_{tailor} \nabla_\theta \mathcal{L}^{\text{tailor}}(\theta, x_i)$     `// Inner step with tailor loss`
   $\theta = \theta - \lambda_{sup} \nabla_\theta \sum_{(x_i, y_i)} \mathcal{L}^{\text{sup}}\left(f_{\theta_{x_i}}(x_i), y_i\right)$   `// Outer step with supervised loss`
  **return** $\theta$

---

tomized parameters $\theta_x$ and use them to make the final prediction: $f_{\theta_x}(x)$. We call this process *tailoring*, because we adapt the model to each particular input for a customized fit. Notice that tailoring optimizes the loss at the query input, eliminating the generalization gap on the unsupervised auxiliary loss.

**Meta-tailoring** Since we will be applying tailoring at prediction time, it is natural to incorporate this adaptation during training, resulting in a two-layer optimization similar to those used in meta-learning. Because of this similarity, we call this process *meta-tailoring*, illustrated in the bottom row of Figure 1. Now, rather than letting $\hat{\theta}$ be the direct minimizer of the supervised loss, we set it to

$$\hat{\theta} \in \arg\min_\theta \sum_{i=1}^n \mathcal{L}^{\text{sup}}(f_{\tau(\theta, \mathcal{L}^{\text{tailor}}, x_i)}(x_i), y_i).$$

Here, the inner loop optimizes the unsupervised tailoring loss $\mathcal{L}^{\text{tailor}}$ and the outer loop optimizes the supervised task loss $\mathcal{L}^{\text{sup}}$. Notice that now the outer process optimizes the only objective we care, $\mathcal{L}^{\text{sup}}$, instead of a proxy combination of $\mathcal{L}^{\text{sup}}$ and $\mathcal{L}^{\text{unsup}}$. At the same time, we learn to leverage $\mathcal{L}^{\text{tailor}}$ in the inner loop to affect the model before making the final prediction, both during training and evaluation. Adaptation is especially clear in the case of a single gradient step, as in MAML [19]. We show its translation, MAMmoTh (Model-Agnostic Meta-Tailoring), in algorithm 1.

In many settings, we want to make predictions for a large number of queries in a (mini-)batch. While MAMmoTh adapts to every input separately, it can only be run efficiently in parallel in some deep learning frameworks, such as JAX [10]. Inspired by conditional normalization (CN) [18] we propose CNGRAD, which adds element-wise affine transformations to our model and only adapts the added parameters in the inner loop. This allows us to independently *tailor* the model for multiple inputs in parallel. We prove theoretically, in Sec. 4, and provide experimental evidence, in Sec. 5.1, that optimizing these parameters alone has enough capacity to minimize a large class of tailoring losses.

**Relation between (meta-)tailoring, fine-tuning transfer, and meta-learning** Fine-tuning pretrained networks is a fruitful method of transferring knowledge from large corpora to smaller related datasets [17]. This allows us to reuse features on related tasks or for different distributions of the same task. When the data we want to adapt to is unlabeled, we must use unsupervised losses. This can be useful to adapt to changes of task [16], from simulated to real data [52], or to new distributions [46].

Tailoring performs unsupervised fine-tuning and is, in this sense, similar to test-time training(TTT) [46] for a single sample, which adapts to distribution shifts. However, tailoring is applied to a single query; not to a data set that captures distribution shift, where batched TTT sees most of its benefits. Thus, whereas regular fine-tuning benefits from more adaptation data, tailoring would be hindered by adapting simultaneously to more data. This is because tailoring aims at building a custom model for each query to ensure the network satisfies a particular inductive bias. Customizing the model to multiple samples makes it harder, not easier. We show this in Figure 2, where TTT with 6400 samples performs worse than tailoring with a single sample. Furthermore, tailoring adapts to each query one by one, not globally from training data to test data. Therefore, it also makes sense to do tailoring on training queries (i.e., meta-tailoring).

Meta-tailoring has the same two-layer optimization structure as meta-learning. More concretely, it can be understood as the extreme case of meta-learning where each single-query prediction is its own task. However, whereas meta-learning tasks use one loss and different examples for the inner and outer loop, meta-tailoring tasks use one example and different losses for each loop ($\mathcal{L}^{\text{tailor}}, \mathcal{L}^{\text{sup}}$). We emphasize that meta-tailoring does not operate in the typical multi-task meta-learning setting. Instead, we are leveraging techniques from meta-learning for the classical single-task setting.

**Contributions** In summary, our contributions are:

1. Introducing *tailoring*, a new framework for encoding inductive biases by minimizing unsupervised losses at prediction time, with theoretical guarantees and broad potential applications.

2. Formulating *meta-tailoring*, which adjusts the outer objective to optimize only the task loss, and developing a new algorithm, CNGRAD, for efficient meta-tailoring.

3. Demonstrating *meta-tailoring* in 3 domains: encoding hard and soft conservation laws in physics prediction problems (Sec. 5.1 and Sec. 5.2), enhancing resistance to adversarial examples by increasing local smoothness at prediction time (Sec. 5.4), and improving prediction quality both theoretically (Sec. 3.1) and empirically (Sec. 5.3) by tailoring with a contrastive loss.

## 2 Related work

Tailoring is inspired by transductive learning. However, transductive methods, because they operate on a batch of unlabeled queries, are allowed to make use of the underlying distributional properties of those queries, as in semi-supervised learning [12]. In contrast, tailoring does the bulk of the computations before receiving any query; vastly increasing efficiency. Similar to tailoring, local learning [9] also has input-dependent parameters. However, it uses similarity in raw input space to select a few labeled data points and builds a local model instead of reusing the global prior learned across the whole data. Finally, some methods [21, 33] in meta-learning propagate predictions along the test samples in a semi-supervised transductive fashion.

Similar to tailoring, there are other learning frameworks that perform optimization at prediction time for very different purposes. Among those, energy-based models do generative modeling [2, 27, 32] by optimizing the hidden activations of neural networks, and other models [4, 49] learn to solve optimization problems by embedding optimization layers in neural networks. In contrast, tailoring optimizes the parameters of the model, not the hidden activations or the output.

As discussed in the introduction, unsupervised fine-tuning methods have been proposed to adapt to different types of variations between training and testing. Sun et al. [46] propose to adapt to a change of distribution with few samples by unsupervised fine-tuning at test-time, applying it with a loss of predicting whether the input has been rotated. Zhang et al. [54] build on it to adapt to group distribution shifts with a learned loss. Other methods in the few-shot meta-learning setting exploit test samples of a new task by minimizing either entropy [16] or a learned loss [5] in the inner optimization. Finally, Wang et al. [51] use entropy in the inner optimization to adapt to large-scale variations in image segmentation. In contrast, we propose (meta-)tailoring as a general effective way to impose inductive biases in the classic machine learning setting. Whereas in the aforementioned methods, adaptation happens from training to testing, we independently adapt to every single query.

Meta-learning [44, 7, 48, 28] has the same two-level optimization structure as meta-tailoring but focuses on multiple prediction tasks. As shown in Alg. 1 for MAML [19], most optimization-based meta-learning algorithms can be converted to meta-tailoring. Similar to CNGRAD, there are other meta-learning methods whose adaptations can be batched [40, 3]. Among these, [55, 41] train FiLM networks [39] to predict custom conditional normalization (CN) layers for each task. By optimizing the CN layers directly, CNGRAD is simpler, while remaining provably expressive (section 4). CNGrad can also start from a trained model by initializing the CN layers to the identity function.

## 3 Theoretical motivations of meta-tailoring

In this section, we study the potential advantages of meta-tailoring from the theoretical viewpoint, formalizing the intuitions conveyed in the introduction. By acting symmetrically during training and prediction time, meta-tailoring allows us to closely relate its training and expected losses, whereas tailoring alone does not have the same guarantees. First, we analyze the particular case of a contrastive tailoring loss. Then, we will generalize the guarantees to other types of tailoring losses.

### 3.1 Meta-tailoring with a contrastive tailoring loss

*Contrastive learning* [24] has seen significant successes in problems of semi-supervised learning [37, 26, 13]. The main idea is to create multiple versions of each training image and learn a representation in which variations of the same image are close while variations of different images are far apart. Typical augmentations involve cropping, color distortions, and rotation. We show theoretically that, under reasonable conditions, meta-tailoring using a particular contrastive loss $\mathcal{L}_{\text{cont}}$ as $\mathcal{L}^{\text{tailor}} = \mathcal{L}_{\text{cont}}$ helps us improve generalization errors in expectation compared with performing classical inductive learning.

When using meta-tailoring, we define $\theta_{x,S}$ to be the $\theta_x$ obtained with a training dataset $S = ((x_i, y_i))_{i=1}^n$ and tailored with the contrastive loss at the prediction point $x$. Theorem 1 provides an upper bound on the expected supervised loss $\mathbb{E}_{x,y}[\mathcal{L}^{\text{sup}}(f_{\theta_{x,S}}(x), y)]$ in terms of the expected contrastive loss $\mathbb{E}_x[\mathcal{L}_{\text{cont}}(x, \theta_{x,S})]$ (analyzed in App. B), the empirical supervised loss $\frac{1}{n}\sum_{i=1}^n \mathcal{L}^{\text{sup}}(f_{\theta_{x_i,S}}(x_i), y_i)$ of meta-tailoring, and its uniform stability $\zeta$. Theorem 6 (App. C) provides a similar bound with the Rademacher complexity [6] $\mathcal{R}_n(\mathcal{L}^{\text{sup}} \circ \mathcal{F})$ of the set $\mathcal{L}^{\text{sup}} \circ \mathcal{F}$, instead of using the uniform stability $\zeta$. Proofs of all results in this paper are deferred to App. C.

**Definition 1.** Let $S = ((x_i, y_i))_{i=1}^n$ and $S' = ((x_i', y_i'))_{i=1}^n$ be any two training datasets that differ by a single point. Then, a meta-tailoring algorithm $S \mapsto f_{\theta_{\mathbf{x},S}}(x)$ is *uniformly $\zeta$-stable* if
$$\forall(x,y) \in \mathcal{X} \times \mathcal{Y}, \ |\mathcal{L}^{\text{sup}}(f_{\theta_{\mathbf{x},S}}(x), y) - \mathcal{L}^{\text{sup}}(f_{\theta_{\mathbf{x},S'}}(x), y)| \leq \frac{\zeta}{n}.$$

**Theorem 1.** *Let $S \mapsto f_{\theta_{\mathbf{x},S}}(x)$ be a uniformly $\zeta$-stable meta-tailoring algorithm. Then, for any $\delta > 0$, with probability at least $1 - \delta$ over an i.i.d. draw of $n$ i.i.d. samples $S = ((x_i, y_i))_{i=1}^n$, the following holds: for any $\kappa \in [0,1]$, $\mathbb{E}_{x,y}[\mathcal{L}^{\text{sup}}(f_{\theta_{\mathbf{x},S}}(x), y)] \leq \kappa \mathbb{E}_x\left[\mathcal{L}_{\text{cont}}(x, \theta_{\mathbf{x},S})\right] + (1-\kappa)\mathcal{J}$, where $\mathcal{J} = \frac{1}{n}\sum_{i=1}^n \mathcal{L}^{\text{sup}}(f_{\theta_{x_i,S}}(x_i), y_i) + \frac{\zeta}{n} + (2\zeta + c)\sqrt{(\ln(1/\delta))/(2n)}$, and $c$ is the upper bound on the per-sample loss as $\mathcal{L}^{\text{sup}}(f_\theta(x), y) \leq c$.*

In the case of regular inductive learning, we get a bound of the exact same form, except that we have a single $\theta$ instead of a $\theta_{\mathbf{x}}$ tailored to each input $x$. This theorem illustrates the effect of meta-tailoring on contrastive learning, with its potential reduction of the expected contrastive loss $\mathbb{E}_x[\mathcal{L}_{\text{cont}}(x, \theta_{x,S})]$. In classic induction, we may aim to minimize the empirical contrastive loss $\frac{1}{\bar{n}}\sum_{i=1}^{\bar{n}} \mathcal{L}_{\text{cont}}(x_i, \theta)$ with $\bar{n}$ potentially unlabeled training samples, which incurs the additional generalization error of $\mathbb{E}_x[\mathcal{L}_{\text{cont}}(x, \theta_{x,S})] - \frac{1}{\bar{n}}\sum_{i=1}^{\bar{n}} \mathcal{L}_{\text{cont}}(x_i, \theta)$. In contrast, meta-tailoring can avoid this extra generalization error by directly minimizing a custom $\theta_{\mathbf{x}}$ on each $x$: $\mathbb{E}_x[\mathcal{L}_{\text{cont}}(x, \theta_{\mathbf{x},S})]$.

In the case where $\mathbb{E}_x[\mathcal{L}_{\text{cont}}(x, \theta_{\mathbf{x},S})]$ is left large (e.g., due to large computational cost), Theorem 1 still illustrates competitive generalization bounds of meta-tailoring with small $\kappa$. For example, with $\kappa = 0$, it provides generalization bounds with the uniform stability for meta-tailoring algorithms. Even then, the bounds are not equivalent to those of classic induction, and there are potential benefits of meta-tailoring, which are discussed in the following section with a more general setting.

### 3.2 Meta-tailoring with general tailoring losses

The benefits of meta-tailoring go beyond contrastive learning: below we provide guarantees for meta-tailoring with arbitrary pairs of tailoring loss $\mathcal{L}^{\text{tailor}}(x, \theta)$ and supervised loss $\mathcal{L}^{\text{sup}}(f_\theta(x), y)$.

**Remark 1.** *For any function $\varphi$ such that $\mathbb{E}_{x,y}[\mathcal{L}^{\text{sup}}(f_\theta(x), y)] \leq \mathbb{E}_x[\varphi(\mathcal{L}^{\text{tailor}}(x, \theta))]$, Theorems 1 and 6 hold with the map $\mathcal{L}_{\text{cont}}$ being replaced by the function $\varphi \circ \mathcal{L}^{\text{tailor}}$.*

This remark shows the benefits of meta-tailoring through its effects on three factors: the expected unlabeled loss $\mathbb{E}_x[\varphi(\mathcal{L}^{\text{tailor}}(x, \theta_{\mathbf{x},S}))]$, uniform stability $\zeta$, and the Rademacher complexity $\mathcal{R}_n(\mathcal{L}^{\text{sup}} \circ \mathcal{F})$. It is important to note that meta-tailoring can directly minimize the expected unlabeled loss $\mathbb{E}_x[\varphi(\mathcal{L}^{\text{tailor}}(x, \theta_{\mathbf{x},S}))]$, whereas classic induction can only minimize its empirical version, which results in the additional generalization error on the difference between the expected unlabeled loss and its empirical version. For example, if $\varphi$ is monotonically increasing and $\mathcal{L}^{\text{tailor}}(x, \theta)$ represents the physical constraints at each input $x$ (as in the application in section 5.1), then classic induction requires a neural network trained to conserve energy at the *training* points to generalize to also conserve it at *unseen* (e.g., testing) points. Meta-tailoring avoids this requirement by directly minimizing violations of energy conservation at each point at prediction time.

Meta-tailoring can also improve the *parameter stability* $\zeta_\theta$ defined such that $\forall(x,y) \in \mathcal{X} \times \mathcal{Y}, \|\theta_{x,S} - \theta_{x,S'}\| \leq \frac{\zeta_\theta}{n}$, for all $S, S'$ differing by a single point. When $\theta_{x,S} = \hat{\theta}_S - \lambda\nabla\mathcal{L}^{\text{tailor}}(x, \hat{\theta}_S)$, we obtain an improvement on the parameter stability $\zeta_\theta$ if $\nabla\mathcal{L}^{\text{tailor}}(x, \hat{\theta}_S)$ can pull $\hat{\theta}_S$ and $\hat{\theta}_{S'}$ closer so that $\|\theta_{x,S} - \theta_{x,S'}\| < \|\hat{\theta}_S - \hat{\theta}_{S'}\|$, which is ensured, for example, if $\|\cdot\| = \|\cdot\|_2$ and $\text{cos\_dist}(v_1, v_2)\frac{\|v_1\|}{\|v_2\|} > \frac{1}{2}$ where $\text{cos\_dist}(v_1, v_2)$ is the cosine similarity of $v_1$ and $v_2$, with $v_1 = \hat{\theta}_S - \hat{\theta}_{S'}, v_2 = \lambda(\nabla\mathcal{L}^{\text{tailor}}(x, \hat{\theta}_S) - \nabla\mathcal{L}^{\text{tailor}}(x, \hat{\theta}_{S'}))$ and $v_2 \neq 0$. Here, the uniform stability $\zeta$ and the parameter stability $\zeta_\theta$ are closely related as $\zeta \leq C\zeta_\theta$, where $C$ is the upper bound on the Lipschitz constants of the maps $\theta \mapsto \mathcal{L}^{\text{sup}}(f_\theta(x), y)$ over all $(x, y) \in \mathcal{X} \times \mathcal{Y}$ under the norm $\|\cdot\|$, since $|\mathcal{L}^{\text{sup}}(f_{\theta_{x,S}}(x), y) - \mathcal{L}^{\text{sup}}(f_{\theta_{x,S'}}(x), y)| \leq C\|\theta_{x,S} - \theta_{x,S'}\| \leq \frac{C\zeta_\theta}{n}$.

**Algorithm 2** CNGRAD for meta-tailoring

---

**Subroutine** *Training($f$, $\mathcal{L}^{\text{sup}}$, $\lambda_{sup}$, $\mathcal{L}^{\text{tailor}}$, $\lambda_{tailor}$, $steps$, $\mathcal{D}_{train}$, $b$)* `// Only in meta-tailoring`
   randomly initialize $w$      `// All parameters except` $\gamma, \beta$`; trained in outer loop`
   **while** *not done* **do**
     $X, Y \sim^b \mathcal{D}_{train}; grad_w = 0$      `// Sample batch; initialize outer gradient`
     $\gamma_0 = \mathbf{1}_{b, \sum_l m_l}; \beta_0 = \mathbf{0}_{b, \sum_l m_l}$      `// Initialize CN layers to the identity`
     **for** $1 \leq s \leq steps$ **do**
       $\gamma_s = \gamma_{s-1} - \lambda_{tailor} \nabla_\gamma \mathcal{L}^{\text{tailor}}(w, \gamma_{s-1}, \beta_{s-1}, X)$      `// Inner step w.r.t.` $\gamma$
       $\beta_s = \beta_{s-1} - \lambda_{tailor} \nabla_\beta \mathcal{L}^{\text{tailor}}(w, \gamma_{s-1}, \beta_{s-1}, X)$      `// Inner step w.r.t.` $\beta$
       $\gamma_s, \beta_s = \gamma_s.detach(), \beta_s.detach()$      `// Only in` $1^{st}$ `order CNGrad`
       $grad_w = grad_w + \nabla_w \mathcal{L}^{\text{sup}}(f_{w, \gamma_s, \beta_s}(X), Y)$      `// Outer gradient w.r.t.` $w$
     $w = w - \lambda_{sup} grad_w$      `// Apply outer step after all inner steps`
   **return** $w$

**Subroutine** *Prediction($f$, $w$, $\mathcal{L}^{\text{tailor}}$, $\lambda$, $steps$, $X$)* `// Both in meta-tailoring & tailoring`
   $\gamma_0 = \mathbf{1}_{X.shape[0], \sum_l m_l}; \beta_0 = \mathbf{0}_{X.shape[0], \sum_l m_l}$
   **for** $1 \leq s \leq steps$ **do**
     $\gamma_s = \gamma_{s-1} - \lambda \nabla_\gamma \mathcal{L}^{\text{tailor}}(w, \gamma_{s-1}, \beta_{s-1}, X)$
     $\beta_s = \beta_{s-1} - \lambda \nabla_\beta \mathcal{L}^{\text{tailor}}(w, \gamma_{s-1}, \beta_{s-1}, X)$
   **return** $f_{w, \gamma_{steps}, \beta_{steps}}(X)$

---

## 4 CNGRAD: a simple algorithm for expressive, efficient (meta-)tailoring

In this section, we address the issue of using (meta-)tailoring for efficient GPU computations. Although possible in JAX [10], efficiently parallelizing MAMmoTh across inputs is not possible in other frameworks. To overcome this issue, building on CAVIA [55] and WarpGrad [20], we propose CNGRAD which adapts only *conditional normalization* parameters and enables efficient GPU computations for (meta-)tailoring. CNGRAD can also be used in meta-learning, providing a parallelizable alternative to MAML (see App. D).

As done in batch-norm [30] after element-wise normalization, we can implement an element-wise affine transformation with parameters $(\gamma, \beta)$, scaling and shifting the output $h_k^{(l)}(x)$ of each $k$-th neuron at the $l$-th hidden layer independently: $\gamma_k^{(l)} h_k^{(l)}(x) + \beta_k^{(l)}$. In conditional normalization, Dumoulin et al. [18] train a collection of $(\gamma, \beta)$ in a multi-task fashion to learn different tasks with a single network. CNGRAD brings this concept to the meta-learning and (meta-)tailoring settings and adapts the affine parameters $(\gamma, \beta)$ to each query. For meta-tailoring, the inner loop minimizes the tailoring loss at an input $x$ by adjusting the affine parameters and the outer optimization adapts the rest of the network. Similar to MAML [19], we implement a first-order version, which does not backpropagate through the optimization, and a second-order version, which does. CNGRAD efficiently parallelizes computations of multiple tailored models because the adapted parameters only require element-wise multiplications and additions. See Alg. 2 for the pseudo-code.

CNGRAD is widely applicable since the adaptable affine parameters can be added to any hidden layer and only represent a tiny portion of the network (empirically, around $1\%$). Moreover, we can see that, under realistic assumptions, we can minimize the inner tailoring loss using only the affine parameters. To analyze properties of these adaptable affine parameters, let us decompose $\theta$ into $\theta = (w, \gamma, \beta)$, where $w$ contains all the weight parameters (including bias terms), and the $(\gamma, \beta)$ contains all the affine parameters. Given an arbitrary function $(f_\theta(x), x) \mapsto \ell_{\text{tailor}}(f_\theta(x), x)$, let $\mathcal{L}^{\text{tailor}}(x, \theta) = \sum_{i=1}^{n_g} \ell_{\text{tailor}}(f_\theta(g^{(i)}(x)), x)$, where $g^{(1):(n_g)}$ are arbitrary input augmentation functions at prediction time.

Corollary 1 states that for any given $\hat{w}$, if we add any non-degenerate Gaussian noise $\delta$ as $\hat{w} + \delta$ with zero mean and any variance on $\delta$, the global minimum value of $\mathcal{L}^{\text{tailor}}$ w.r.t. all parameters $(w, \gamma, \beta)$ can be achieved by optimizing only the affine parameters $(\gamma, \beta)$, with probability one. In other words, the CN parameters $(\gamma, \beta)$ have enough capacity to optimize optimize the inner tailoring loss.

**Corollary 1.** *Under the assumptions of Theorem 2, for any $\hat{w} \in \mathbb{R}^d$, with probability one over randomly sampled $\delta \in \mathbb{R}^d$ accordingly to any non-degenerate Gaussian distribution, the following holds:* $\inf_{w, \gamma, \beta} \mathcal{L}^{\text{tailor}}(x, w, \gamma, \beta) = \inf_{\gamma, \beta} \mathcal{L}^{\text{tailor}}(x, \hat{w} + \delta, \gamma, \beta)$ *for any $x \in \mathcal{X}$.*

The assumption and condition in theorem 2 are satisfied in practice (see App. A). Therefore, CNGRAD is a practical and computationally efficient method to implement (meta-)tailoring.

| Method | loss | relative |
|---|---|---|
| Inductive learning | .041 | - |
| Opt. output(50 st.) | .041 | $(0.7 \pm 0.1)\%$ |
| 6400-s. TTT(50 st.) | .040 | $(3.6 \pm 0.2)\%$ |
| Tailoring(1 step) | .040 | $(1.9 \pm 0.2)\%$ |
| Tailoring(5 steps) | .039 | $(6.3 \pm 0.3)\%$ |
| Tailoring(10 st.) | .038 | $(7.5 \pm 0.1)\%$ |
| Meta-tailoring(0 st.) | .030 | $(26.3 \pm 3.3)\%$ |
| Meta-tailoring(1 st.) | .029 | $(29.9 \pm 3.0)\%$ |
| Meta-tailoring(5 st.) | .027 | $(35.3 \pm 2.6)\%$ |
| Meta-tailoring(10 s.) | .026 | $(36.0 \pm 2.6)\%$ |

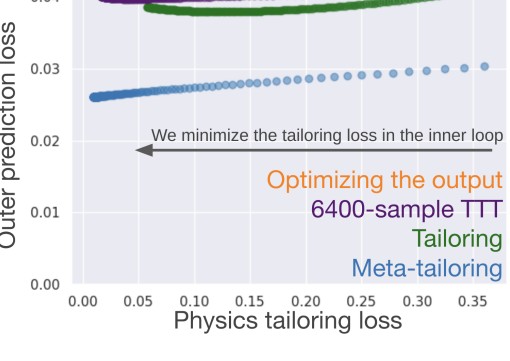

Table 1: Test MSE loss for different methods; the second column shows the relative improvement over basic inductive supervised learning. The test-time training (TTT) baseline uses a full batch of 6400 test samples to adapt, not allowed in regular SL. With a few gradient steps, tailoring significantly over-performs all baselines. Meta-tailoring improves even further, with $35\%$ improvement.

Figure 2: Optimization at prediction time on the planet data; each path going from right to left as we minimize the physics tailoring loss. We use a small step size to illustrate the path. Tailoring and the two baselines only differ in their test-time computations, thus sharing their starts. Meta-tailoring has a lower starting loss, faster optimization, and no overfitting during tailoring.

## 5 Experiments

### 5.1 Tailoring to impose symmetries and constraints at prediction time

Exploiting invariances and symmetries is an established strategy for increasing performance in ML. During training, we can regularize networks to satisfy specific criteria; but this does not guarantee they will be satisfied outside the training dataset [45]. (Meta-)tailoring provides a general solution to this problem by adapting the model to satisfy the criteria at prediction time. We demonstrate the use of tailoring to enforce physical conservation laws for predicting the evolution of a 5-body planetary system. This prediction problem is challenging, as $m$-body systems become chaotic for $m > 2$. We generate a dataset with positions, velocities, and masses of all 5 bodies as inputs and the changes in position and velocity as targets. App. E further describes the dataset.

Our model is a 3-layer feed-forward network. We tailor it by taking the original predictions and adapting the model using the tailoring loss given by the $L_1$ loss between the whole system's initial and final energy and momentum. Note that ensuring this conservation does not guarantee better performance: predicting the input as the output conserves energy and momentum perfectly, but it is not correct.

While tailoring adapts some parameters in the network to improve the tailoring loss, an alternative for enforcing conservation would be to adapt the output $y$ value directly. Table 1 compares the predictive accuracy of inductive learning, direct output optimization, and both tailoring and meta-tailoring, using varying numbers of gradient steps. Tailoring is more effective than adapting the output, as the parameters provide a prior on what changes are more natural. For meta-tailoring, we try both first-order and second-order versions of CNGRAD. The first-order gave slightly better results, possibly because it was trained with a higher tailor learning rate ($10^{-3}$) with which the second-order version was unstable (we thus used $10^{-4}$). More details can be found in App. E.

Finally, meta-tailoring without any query-time tailoring steps already performs much better than the original model, even though both have almost the same number of parameters and can overfit the dataset. We conjecture meta-tailoring training adds an inductive bias that guides optimization towards learning a more generalizable model. Fig. 2 shows prediction-time optimization paths.

### 5.2 Tailoring to softly encourage inductive biases

A popular way of encoding inductive biases is with clever network design to make predictions translation equivariant (CNNs), permutation equivariant (GNNs), or conserve energy [23]. However, if an inductive bias is only partially satisfied, such approaches overly constrain the function class. Instead, tailoring can softly impose this bias by only fine-tuning the tailoring loss for a few steps.

We showcase this in the real pendulum experiment used by Hamiltonian Neural Networks (HNNs) [23]. HNNs have energy conservation built-in and easily improve a vanilla MLP. We meta-tailor this vanilla MLP with energy conservation without changing its architecture. Meta-tailoring significantly improves over the baseline and HNNs, since it can encode the *imperfect* energy conservation of real systems. We compare results in Fig. 3 and provide extra details in App. F. Note that, with inexact losses, fully enforcing them provides

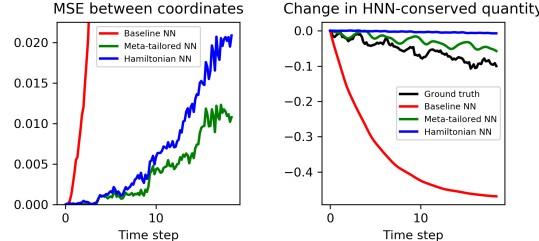

Figure 3: By softly encouraging energy conservation, meta-tailoring improves over models that don't and models that fully impose it.

sub-optimal results. Thus, we pick the tailoring learning rate that results in the lowest long-term prediction loss during training.

## 5.3 Tailoring with a contrastive loss for image classification

Following the setting described in section 3.2, we provide experiments on the CIFAR-10 dataset [31] by building on SimCLR [13]. SimCLR trains a ResNet-50 [25] $f_\theta(\cdot)$ coupled to a small MLP $g(\cdot)$ such that the outputs of two augmentations of the same image $x_i, x_j \sim \mathcal{T}(x)$ agree; i.e. $g(f_\theta(x_i)) \approx g(f_\theta(x_j))$. This is done by training $g(f(\cdot))$ to recognize one augmentation from the other among a big batch of candidates with the cross-entropy loss. To show that the unsupervised training of $f_\theta$ provides a useful representation, SimCLR trains a single linear layer on top of it, $\phi(f_\theta(\cdot))$, achieving good classification results.

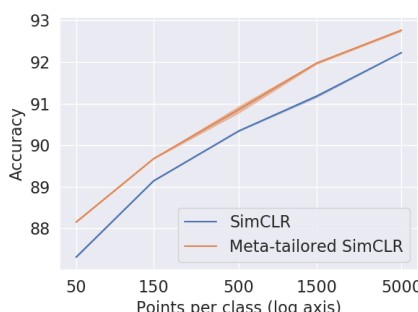

We now observe that we can tailor $f_\theta$ at prediction-time by optimizing $g(f_{\theta_x}(x))$, which quantifies the agreement between different augmentations of the same input; thus 'learning' about its particularities. To make the image classification prediction, we feed the final tailored representation to the linear layer: $\phi(f_{\theta_x}(x))$. To match the evaluation from SimCLR, we do not redo SimCLR's un-

Figure 4: Meta-tailoring the linear layer with the contrastive loss results in consistent accuracy gains between $0.5\%$ and $0.8\%$. This is approximately the same gain as that of doubling the amount of labeled data (note the logarithmic x-axis).

supervised learning, which provides $\theta$. The meta-tailoring outer loop trains $\phi$ to take the tailored representations $f_{\theta_x}(x)$ instead of the original $f_\theta(x)$. Thus, $\theta$ is unsupervisedly fine-tuned in the prediction function leading to $\theta_x$, but never supervisedly trained as this would break the evaluation protocol (in meta-tailoring's favor). We also implement a TTT [46] baseline with their original rotation-prediction loss. Moreover, TTT modifies $\theta_x$ at test time, but does not take this adaptation into account when training $\phi$ (see App. G for more details). TTT worsened base SimCLR despite significant hyper-parameter tuning. We conjecture this is because TTT was designed for OOD generalization, not in-distribution. In contrast, as shown in Fig. 4, we observe that meta-tailoring provides improvements over base SimCLR equivalent to doubling the amount of labeled data.

## 5.4 Tailoring for robustness against adversarial examples

Neural networks are susceptible to adversarial examples [8, 47]: targeted small perturbations of an input can cause the network to misclassify it. One approach is to make the prediction function smooth via adversarial training [34]; however, this only ensures smoothness in the training points. Constraining the model to be smooth everywhere makes it lose capacity. Instead, (meta-)tailoring asks for smoothness *a posteriori*, only on a specific query.

We apply meta-tailoring to robustly classifying CIFAR-10 [31] and ImageNet [15] images, tailoring predictions so that they are locally smooth. This is similar to VAT [36] but instead optimizes the loss within the prediction function, not as an auxiliary loss. Inspired by the notion of adversarial examples being caused by predictive, but non-robust, features [29], we meta-tailor our model by enforcing smoothness on the vector of features of the penultimate layer (denoted $g_\theta(x)$):

$$\mathcal{L}^{\text{tailor}}(x, \theta) = \mathbb{E}[\text{cos\_dist}(g_\theta(x), g_\theta(x + \delta))], \delta \sim N(0, \nu^2),$$

| $\sigma$ | Method | 0.0 | 0.5 | 1.0 | 1.5 | 2.0 | 2.5 | 3.0 | ACR |
|---|---|---|---|---|---|---|---|---|---|
| 0.25 | (Inductive) Randomized Smoothing | 0.67 | 0.49 | 0.00 | 0.00 | 0.00 | 0.00 | 0.00 | 0.470 |
| | Meta-tailored Randomized Smoothing | **0.72** | **0.55** | 0.00 | 0.00 | 0.00 | 0.00 | 0.00 | **0.494** |
| 0.50 | (Inductive) Randomized Smoothing | 0.57 | 0.46 | 0.37 | 0.29 | 0.00 | 0.00 | 0.00 | 0.720 |
| | Meta-tailored Randomized Smoothing | 0.66 | 0.54 | **0.42** | **0.31** | 0.00 | 0.00 | 0.00 | **0.819** |
| 1.00 | (Inductive) Randomized Smoothing | 0.44 | 0.38 | 0.33 | 0.26 | 0.19 | 0.15 | 0.12 | 0.863 |
| | Meta-tailored Randomized Smoothing | 0.52 | 0.45 | 0.36 | **0.31** | **0.24** | **0.20** | **0.15** | **1.032** |

Table 2: Fraction of points with certificate above different radii for ImageNet. Meta-tailoring improves average certification radius (ACR) of different models by $5.1\%, 13.8\%, 19.6\%$. Results for Randomized Smoothing come from [53].

We build on Cohen et al. [14], who developed a method for certifying the robustness of a model via randomized smoothing (RS). RS samples points from a Gaussian $N(x, \sigma^2)$ around the query and, if there is enough agreement in classification, it provides a certificate that a small perturbation cannot adversarially modify the query to have a different class. We show that meta-tailoring improves the original RS method, testing for $\sigma = 0.25, 0.5, 1.0$. We use $\nu = 0.1$ for all experiments. We initialized with the weights of Cohen et al. [14] by leveraging that CNGRAD can start from a pre-trained model by initializing the extra affine layers to the identity. Finally, we use $\sigma' = \sqrt{\sigma^2 - \nu^2} \approx 0.23, 0.49, 0.995$ so that the points used in our tailoring loss come from $N(x, \sigma^2)$.

Table 7 shows our results on CIFAR-10 where we improve the average certification radius (ARC) by $8.6\%, 10.4\%, 19.2\%$ respectively. In table 2, we show results on Imagenet where we improve the ARC by $5.1\%, 13.8\%, 19.6\%$ respectively. We chose to meta-tailor the RS method because it represents a strong standard in certified adversarial defenses, but we note that there have been advances on RS that sometimes achieve better results than those presented here [53, 43], see App. I. However, it is likely that meta-tailoring could also improve these methods.

These experiments only scratch the surface of what tailoring allows for adversarial defenses: usually, the adversary looks at the model and gets to pick a particularly bad perturbation $x + \delta$. With tailoring, the model responds, by changing to weights $\theta_{x+\delta}$. This leads to a game, where both weights and inputs are perturbed, similar to $\max_{|\delta| < \epsilon_x} \min_{|\Delta| < \epsilon_\theta} \mathcal{L}^{\sup}(f_{\theta+\Delta}(x + \delta), y)$. However, since we don't get to observe $y$; we optimize the weight perturbation by minimizing $\mathcal{L}^{\text{tailor}}$ instead.

## 6 Discussion

### 6.1 Broader Impact

Improving adversarial robustness: having more robust and secure ML systems is mostly a positive change. However, improving adversarial defenses could also go against privacy preservation, like the use of adversarial patches to gain anonymity from facial recognition. Encoding desirable properties: By optimizing an unsupervised loss for the particular query we care about, it is easier to have guarantees on the prediction. In particular, there could be potential applications for fairness, where the unsupervised objective could enforce specific criteria at the query or related inputs. More research needs to be done to make this assertion formal and practical. Potential effect on privacy: tailoring specializes the model to each input. This could have an impact on privacy. Intuitively, the untailored model can be less specialized to each input, lowering the individual information from each training point contained in the model. However, tailored predictions extract more information about the queries, from which more personal information could be leaked.

### 6.2 Limitations

Tailoring provides a framework for encoding a wide array of inductive biases, but these need to be specified as a formula by the user. For instance, it would be hard to programatically describe tailoring losses in raw pixel data, such as mass conservation in pixel space. Tailoring also incurs an extra time cost at prediction time, since we make an inner optimization inside the prediction function. However, as shown in Table 1, meta-tailoring often achieves better results than inductive learning even without adaptation at test-time, enabling better predictions at regular speed during test-time. This is due to meta-tailoring leading to better training. Moreover, optimization can be sped up by only tailoring the last layers, as discussed in App. D. Finally, to the best of our knowledge using MAMmoTh for meta-tailoring would be hard to parallelize in PyTorch [38] and Tensorflow [1]; we

proposed CNGRAD to make it easy and efficient. JAX[10], which handles per-example weights, makes parallelizing tailoring effortless.

Theory in Sec. 3 applies only to meta-tailoring. Unlike tailoring (and test-time training), meta-tailoring performs the same computations at training and testing time, which allows us to prove the results. Theorem 2 proves that optimizing the CN layers in CNGRAD has the same expressive power as optimizing all the layers for the inner (not outer) loss. However, it does not guarantee that gradient descent will find the appropriate optima. The study of such guarantee is left for future work.

### 6.3 Conclusion

We have presented *tailoring*, a simple way of embedding a powerful class of inductive biases into models, by minimizing unsupervised objectives at prediction time. Tailoring leverages the generality of auxiliary losses and improves them in two ways: first, it eliminates the generalization gap on the auxiliary loss by optimizing it on the query point; second, tailoring only minimizes task loss in the outer optimization and the tailoring loss in the inner optimization. This results in the model optimizing the only objective we care about in the outer loop, instead of a proxy loss. Beyond inductive biases, tailoring shows that model adaptation is useful even when test queries comes from the same distribution as the training data. This suggests one can improve models by performing prediction-time optimization, trading off large offline data&compute efforts with small online computations.

Tailoring is broadly applicable, as one can vary the model, the unsupervised loss, and the task loss. We show its applicability in three diverse domains: physics prediction time-series, contrastive learning, and adversarial robustness. We also provide a simple algorithm, CNGRAD, to make meta-tailoring practical with little additional code. Currently, most unsupervised or self-supervised objectives are optimized in task-agnostic ways; without taking into account the supervised downstream task. Instead, meta-tailoring provides a generic way to make these objectives especially useful for each application. It does so by learning how to best leverage the unsupervised loss to perform well on the final task we care about.

## Acknowledgments and Disclosure of Funding

We would like to thank Kelsey Allen, Marc de la Barrera, Jeremy Cohen, Dylan Doblar, Chelsea Finn, Sebastian Flennerhag, Jiayuan Mao, Josh Tenenbaum, and Shengtong Zhang for insightful discussions. We would also like to thank Clement Gehring for his help with deploying the experiments and Lauren Milechin for her help with leveraging the MIT supercloud platform [42].

We gratefully acknowledge support from NSF grant 1723381; from AFOSR grant FA9550-17-1-0165; from ONR grant N00014-18-1-2847; from the Honda Research Institute, from MIT-IBM Watson Lab; and from SUTD Temasek Laboratories. We also acknowledge the MIT SuperCloud and Lincoln Laboratory Supercomputing Center for providing HPC resources that have contributed to the reported research results. Any opinions, findings, and conclusions or recommendations expressed in this material are those of the authors and do not necessarily reflect the views of our sponsors.

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
