# A  Theorem 2, Corollary 1 and interpretation of their conditions

**Assumption 1.** *(Common activation)* The activation function $\sigma(x)$ is real analytic, monotonically increasing, and the limits exist as: $\lim_{x\to-\infty}\sigma(x) = \sigma_- > -\infty$ and $\lim_{x\to+\infty}\sigma(x) = \sigma_+ \le +\infty$.

**Theorem 2.** *For any $x \in \mathcal{X}$ that satisfies $\|g^{(i)}(x)\|_2^2 - g^{(i)}(x)^\top g^{(j)}(x) > 0$ (for all $i \neq j$), and for any fully-connected neural network with a single output unit, at least $n_g$ neurons per hidden layer, and activation functions that satisfy Assumption 1, the following holds: $\inf_{w,\gamma,\beta}\mathcal{L}^{\text{tailor}}(x,w,\gamma,\beta) = \inf_{\gamma,\beta}\mathcal{L}^{\text{tailor}}(x,\bar{w},\gamma,\beta)$ for any $\bar{w} \notin \mathcal{W}$ where Lebesgue measure of $\mathcal{W} \subset \mathbb{R}^d$ is zero.*

Assumption 1 is satisfied by using common activation functions such as sigmoid and hyperbolic tangent, as well as *softplus*, which is defined as $\sigma_\alpha(x) = \ln(1+\exp(\alpha x))/\alpha$ and satisfies Assumption 1 with any hyperparameter $\alpha \in \mathbb{R}_{>0}$. The softplus activation function can approximate the ReLU function to any desired accuracy: i.e., $\sigma_\alpha(x) \to \text{relu}(x)$ as $\alpha \to \infty$, where $\text{relu}$ represents the ReLU function.

In Theorem 2 and Corollary 1, the condition $\|g^{(i)}(x)\|_2^2 - g^{(i)}(x)^\top g^{(j)}(x) > 0$ (for all $i \neq j$) can be easily satisfied, for example, by choosing $g^{(1)},\ldots,g^{(n_g)}$ to produce normalized and distinguishable argumented inputs for each prediction point $x$ at prediction time. To see this, with normalization $\|g^{(i)}(x)\|_2^2 = \|g^{(j)}(x)\|_2^2$, the condition is satisfied if $\|g^{(i)}(x) - g^{(j)}(x)\|_2^2 > 0$ for $i \neq j$ since $\frac{1}{2}\|g^{(i)}(x) - g^{(j)}(x)\|_2^2 = \|g^{(i)}(x)\|_2^2 - g^{(i)}(x)^\top g^{(j)}(x)$.

In general, the normalization is not necessary for the condition to hold; e.g., orthogonality on $g^{(i)}(x)$ and $g^{(j)}(x)$ along with $g^{(i)}(x) \neq 0$ satisfies it without the normalization.

# B  Understanding the expected meta-tailoring contrastive loss

To analyze meta-tailoring for contrastive learning, we focus on the binary classification loss of the form $\mathcal{L}^{\text{sup}}(f_\theta(x),y) = \ell_{\text{cont}}(f_\theta(x)_y - f_\theta(x)_{y'=\neg y})$ where $\ell_{\text{cont}}$ is convex and $\ell_{\text{cont}}(0) = 1$. With this, the objective function $\theta \mapsto \mathcal{L}^{\text{sup}}(f_\theta(x),y)$ is still non-convex in general. For example, the standard hinge loss $\ell_{\text{cont}}(z) = \max\{0, 1 - z\}$ and the logistic loss $\ell_{\text{cont}}(z) = s\log_2(1 + \exp(z))$ satisfy this condition.

We first define the meta-tailoring contrastive loss $\mathcal{L}_{\text{cont}}(x,\theta)$ in detail. In meta-tailoring contrastive learning, we choose the probability measure of positive example $x^+ \sim \mu_{x^+}(x)$ and the probability measure of negative example $x^-, y^- \sim \mu_{x^-,y^-}(x)$, both of which are tailored for each input point $x$ at prediction time. These choices induce the marginal distributions for the negative examples $y^- \sim \mu_{y^-}(x)$ and $x^- \sim \mu_{x^-}(x)$, as well as the unknown probability of $y^- = y$ defined by $\rho_y(\mu_{y^-}(x)) = \mathbb{E}_{y^-\sim\mu_{y^-}(x)}(\mathbb{1}\{y^- = y\})$. Define the lower and upper bound on the probability of $y^- = y$ as $\underline{\rho}(x) \le \rho_y(\mu_{y^-}(x)) \le \bar{\rho}(x) \in [0,1)$.

Then, the first pre-meta-tailoring contrastive loss can be defined by

$$\mathcal{L}_{\text{cont}}^{x^+,x^-}(x,\theta) = \mathbb{E}_{\substack{x^+\sim\mu_{x^+}(x), \\ x^-\sim\mu_{x^-}(x)}}[\ell_{\text{cont}}(h_\theta(x)^\top(h_\theta(x^+) - h_\theta(x^-)))],$$

where $h_\theta(x) \in \mathbb{R}^{m_H+1}$ represents the output of the last hidden layer, including a constant neuron corresponding the bias term of the last output layer (if there is no bias term, $h_\theta(x) \in \mathbb{R}^{m_H}$). For every $z \in \mathbb{R}^{2\times(m_H+1)}$, define $\psi_{x,y,y^-}(z) = \ell_{\text{cont}}((z_y - z_{y^-})h_\theta(x))$, where $z_y \in \mathbb{R}^{1\times m_H}$ is the $y$-th row vector of $z$. We define the second pre-meta-tailoring contrastive loss by

$$\mathcal{L}_{\text{cont}}^{x^+,x^-,y^-}(x,\theta) = \max_y \mathbb{E}_{y^-\sim\mu_{y^-}(x)}[\psi_{x,y,y^-}(\theta^{(H+1)}) - \psi_{x,1,2}([u_h^+, u_h^-]^\top)],$$

where $u_h^+ = \mathbb{E}_{x^+\sim\mu_{x^+}(x)}[h_\theta(x^+)]$ and $u_h^- = \mathbb{E}_{x^-\sim\mu_{x^-}(x)}[h_\theta(x^-)]$. Here, we decompose $\theta$ as $\theta = (\theta^{(1:H)}, \theta^{(H+1)})$, where $\theta^{(H+1)} = [W^{(H+1)}, b^{(H+1)}] \in \mathbb{R}^{m_y\times(m_H+1)}$ represents the parameters at the last output layer, and $\theta^{(1:H)}$ represents all others.

Then, the meta-tailoring contrastive loss is defined by

$$\mathcal{L}_{\text{cont}}(x,\theta) = \frac{1}{1-\bar{\rho}(x)}\left(\mathcal{L}_{\text{cont}}^{x^+,x^-}(x,\theta) + \mathcal{L}_{\text{cont}}^{x^+,x^-,y^-}(x,\theta) - \underline{\rho}(x)\right).$$

Theorem 3 states that for any $\theta^{(1:H)}$, the convex optimization of $\mathcal{L}_{\text{cont}}^{x^+,x^-}(x,\theta) + \mathcal{L}_{\text{cont}}^{x^+,x^-,y^-}(x,\theta)$ over $\theta^{(H+1)}$ can achieve the value of $\mathcal{L}_{\text{cont}}^{x^+,x^-}(x,\theta)$ without the value of $\mathcal{L}_{\text{cont}}^{x^+,x^-,y^-}(x,\theta)$, allowing us to focus on the first term $\mathcal{L}_{\text{cont}}^{x^+,x^-}(x,\theta)$, for some choice of $\mu_{x^-,y^-}(x)$ and $\mu_{x^+}(x)$.

**Theorem 3.** *For any* $\theta^{(1:H)}, \mu_{x^-,y^-}(x)$ *and* $\mu_{x^+}(x)$, *the function* $\theta^{(H+1)} \mapsto \mathcal{L}_{\text{cont}}^{x^+,x^-}(x,\theta) + \mathcal{L}_{\text{cont}}^{x^+,x^-,y^-}(x,\theta)$ *is convex. Moreover, there exists* $\mu_{x^-,y^-}(x)$ *and* $\mu_{x^+}(x)$ *such that, for any* $\theta^{(1:H)}$ *and any* $\bar{\theta}^{(H+1)}$,

$$\inf_{\theta^{(H+1)} \in \mathbb{R}^{m_y \times (m_H+1)}} \mathcal{L}_{\text{cont}}^{x^+,x^-}(x,\theta) + \mathcal{L}_{\text{cont}}^{x^+,x^-,y^-}(x,\theta) \le \mathcal{L}_{\text{cont}}^{x^+,x^-}(x,\theta^{(1:H)},\bar{\theta}^{(H+1)}).$$

## C Proofs

In order to have concise proofs, we introduce additional notations while keeping track of dependent variables more explicitly. Since $h_\theta$ only depends on $\theta^{(1:H)}$, let us write $h_{\theta^{(1:H)}} = h_\theta$. Similarly, $\mathcal{L}_{\text{cont}}(x,\theta^{(1:H)}) = \mathcal{L}_{\text{cont}}(x,\theta)$. Let $\theta(x) = \theta_x$ and $\theta(x,S) = \theta_{x,S}$. Define $\mathcal{L} = \mathcal{L}^{\text{sup}}$.

### C.1 Proof of Theorem 2

*Proof of Theorem 2.* The output of fully-connected neural networks for an input $x$ with a parameter vector $\theta = (w,\gamma,\beta)$ can be represented by $f_\theta(x) = W^{(H+1)}h^{(H)}(x) + b^{(H+1)}$ where $W^{(H+1)} \in \mathbb{R}^{1 \times m_H}$ and $b^{(H+1)} \in \mathbb{R}$ are the weight matrix and the bias term respectively at the last layer, and $h^{(H)}(x) \in \mathbb{R}^{m_H}$ represents the output of the last hidden layer. Here, $m_l$ represents the number of neurons at the $l$-th layer, and $h^{(l)}(x) = \gamma^{(l)}(\sigma(W^{(l)}h^{(l-1)}(x)+b^{(l)})) - \beta^{(l)} \in \mathbb{R}^{m_l}$ for $l = 1,\ldots,H$, with trainable parameters $\gamma^{(l)},\beta^{(l)} \in \mathbb{R}^{m_l}$, where $h^{(0)}(x) = x$. Let $z^{(l)}(x) = \sigma(W^{(l)}h^{(l-1)}(x) + b^{(l)})$.

Then, by rearranging the definition of the output of the neural networks,

$$f_\theta(x) = W^{(H+1)}h^{(H)}(x) + b^{(H+1)}$$
$$= \left(\sum_{k=1}^{m_H} W_k^{(H+1)}\gamma_k^{(H)}z^{(H)}(x)_k + W_k^{(H+1)}\beta_k^{(H)}\right) + b^{(H+1)}$$
$$= [W^{(H+1)} \circ z^{(H)}(x)^\top, W^{(H+1)}]\begin{bmatrix}\gamma^{(H)}\\\beta^{(H)}\end{bmatrix} + b^{(H+1)}.$$

Thus, we can write

$$\begin{bmatrix}f_\theta(g^{(1)}(x))\\\vdots\\f_\theta(g^{(n_g)}(x))\end{bmatrix} = M_w\begin{bmatrix}\gamma^{(H)}\\\beta^{(H)}\end{bmatrix} + b^{(H+1)}\mathbf{1}_{n_g} \in \mathbb{R}^{n_g}, \tag{1}$$

where

$$M_w = \begin{bmatrix}W^{(H+1)} \circ z^{(H)}(g^{(1)}(x))^\top, W^{(H+1)}\\\vdots\\W^{(H+1)} \circ z^{(H)}(g^{(n_g)}(x))^\top, W^{(H+1)}\end{bmatrix} \in \mathbb{R}^{n_g \times 2m_H},$$

and $\mathbf{1}_{n_g} = [1,1,\ldots,1]^\top \in \mathbb{R}^{n_g}$.

Using the above equality, we show an exitance of a $(\gamma,\beta)$ such that $\mathcal{L}^{\text{tailor}}(x,\bar{w},\gamma,\beta) = \inf_{w,\gamma,\beta}\mathcal{L}^{\text{tailor}}(x,\theta)$ for any $x \in \mathcal{X} \subseteq \mathbb{R}^{m_x}$ and any $\bar{w} \notin \mathcal{W}$ where Lebesgue measure of $\mathcal{W} \subset \mathbb{R}^d$ is zero. To do so, we first fix $\gamma_k^{(l)} = 1$ and $\beta_k^{(l)} = 0$ for $l = 1,\ldots,H-1$, with which $h^{(l)}(x) = z^{(l)}(x)$ for $l = 1,\ldots,H-1$.

Define $\varphi(w) = \det(M_w M_w^\top)$, which is analytic since $\sigma$ is analytic. Furthermore, we have that $\{w \in \mathbb{R}^d : M_w \text{ has rank less than } n_g\} = \{w \in \mathbb{R}^d : \varphi(w) = 0\}$, since the rank of $M_w$ and the rank of the Gram matrix are equal. Since $\varphi$ is analytic, if $\varphi$ is not identically zero ($\varphi \ne 0$), the Lebesgue

measure of its zero set $\{w \in \mathbb{R}^d : \varphi(w) = 0\}$ is zero [35]. Therefore, if $\varphi(w) \neq 0$ for some $w \in \mathbb{R}^d$, the Lebesgue measure of the set $\{w \in \mathbb{R}^d : M_w \text{ has rank less than } n_g\}$ is zero.

Accordingly, we now constructs a $w \in \mathbb{R}^d$ such that $\varphi(w) \neq 0$. Set $W^{(H+1)} = \mathbf{1}_{m_H}^\top$. Then,

$$M_w = [\bar{M}_w, \mathbf{1}_{n_g, m_H}] \in \mathbb{R}^{n_g \times m_H}.$$

where

$$\bar{M}_w = \begin{bmatrix} z^{(H)}(g^{(1)}(x))^\top \\ \vdots \\ z^{(H)}(g^{(n_g)}(x)(x))^\top \end{bmatrix} \in \mathbb{R}^{n_g \times m_H}$$

and $\mathbf{1}_{n_g, m_H} \in \mathbb{R}^{n_g \times m_H}$ with $(\mathbf{1}_{n_g, m_H})_{ij} = 1$ for all $i, j$. For $l = 1, \dots, H$, define

$$G^{(l)} = \begin{bmatrix} z^{(l)}(g^{(1)}(x))^\top \\ \vdots \\ z^{(l)}(g^{(n_g)}(x))^\top \end{bmatrix} \in \mathbb{R}^{n_g \times m_l}.$$

Then, for $l = 1, \dots, H$,

$$G^{(l)} = \sigma(G^{(l-1)}(W^{(l)})^\top + \mathbf{1}_{n_g}(b^{(l)})^\top),$$

where $\sigma$ is applied element-wise (by overloading of the notation $\sigma$), and

$$(\bar{M}_w)_{ik} = (G^{(H)})_{ik}.$$

From the assumption $g(x)$, there exists $c > 0$ such that $\|g^{(i)}(x)\|_2^2 - \langle g^{(i)}(x), g^{(j)}(x) \rangle > c$ for all $i \neq j$. From Assumption 1, there exists $c'$ such that $\sigma_+ - \sigma_- > c'$. Using these constants, set $W_i^{(1)} = \alpha^{(1)} g^{(i)}(x)^\top$ and $b_i^{(1)} = c\alpha^{(1)}/2 - \alpha^{(1)}\|g^{(i)}(x)\|_2^2$ for $i = 1, \dots, n_g$, where $W_i^{(1)}$ represents the $i$-th row of $W^{(1)}$. Moreover, set $W_{1:n_g, 1:n_g}^{(l)} = \alpha^{(l)} I_{n_g}$ and $b_k^{(l)} = c'\alpha^{(l)}/2 - \alpha^{(l)}\sigma_+$ for all $k$ and $l = 2, \dots, H$, where $W_{1:n_g, 1:n_g}^{(l)}$ is the fist $n_g \times n_g$ block matrix of $W^{(1)}$ and $I_{n_g}$ is the $n_g \times n_g$ identity matrix. Set all other weights and bias to be zero. Then, for any $i \in \{1, \dots, n_g\}$,

$$(G^{(1)})_{ii} = \sigma(c\alpha^{(1)}/2),$$

and for any $k \in \{1, \dots, n_g\}$ with $k \neq i$,

$$(G^{(1)})_{ik} = \sigma(\alpha^{(1)}(\langle g^{(i)}(x), g^{(k)}(x) \rangle - \|g^{(k)}(x)\|_2^2 + c/2)) \leq \sigma(-c\alpha^{(1)}/2).$$

Since $\sigma(c\alpha^{(1)}/2) \to \sigma_+$ and $\sigma(-c\alpha^{(1)}/2) \to \sigma_-$ as $\alpha^{(1)} \to \infty$, with $\alpha^{(1)}$ sufficiently large, we have that $\sigma(c\alpha^{(1)}/2) - \sigma_+ + c'/2 \geq c_1^{(2)}$ and $\sigma(-c\alpha^{(1)}/2) - \sigma_+ + c'/2 \leq -c_2^{(2)}$ for some $c_1^{(2)}, c_2^{(2)} > 0$. Note that $c_1^{(2)}$ and $c_2^{(2)}$ depends only on $\alpha^{(1)}$ and does not depend on any of $\alpha^{(2)}, \dots, \alpha^{(H)}$. Therefore, with $\alpha^{(1)}$ sufficiently large,

$$(G^{(2)})_{ii} = \sigma(\alpha^{(2)}(\sigma(c\alpha^{(1)}/2) - \sigma_+ + c'/2)) \geq \sigma(\alpha^{(2)} c_1^{(2)}),$$

and

$$(G^{(2)})_{ik} \leq \sigma(\alpha^{(2)}(\sigma(-c\alpha^{(1)}/2) - \sigma_+ + c'/2)) \leq \sigma(-\alpha^{(2)} c_2^{(2)}).$$

Repeating this process with Assumption 1, we have that with $\alpha^{(1)}, \dots, \alpha^{(H-1)}$ sufficiently large,

$$(G^{(H)})_{ii} \geq \sigma(\alpha^{(H)} c_1^{(H)}),$$

and

$$(G^{(H)})_{ik} \leq \sigma(-\alpha^{(H)} c_2^{(H)}).$$

Here, $(G^{(H)})_{ii} \to \sigma_+$ and $(G^{(H)})_{ik} \to \sigma_-$ as $\alpha^{(H)} \to \infty$. Therefore, with $\alpha^{(1)}, \dots, \alpha^{(H)}$ sufficiently large, for any $i \in \{1, \dots, n_g\}$,

$$\left|(\bar{M}_w)_{ii} - \sigma_-\right| > \sum_{k \neq i} \left|(\bar{M}_w)_{ik} - \sigma_-\right|. \tag{2}$$

The inequality (2) means that the matrix $\bar{M}_w' = [(\bar{M}_w)_{ij} - \sigma_-]_{1 \leq i, j \leq n_g} \in \mathbb{R}^{n_g \times n_g}$ is strictly diagonally dominant and hence is nonsingular with rank $n_g$. This implies that the matrix $[\bar{M}_w', \mathbf{1}_{n_g}] \in$

$\mathbb{R}^{n_g \times (n_g+1)}$ has rank $n_g$. This then implies that the matrix $\tilde{M}_w = [[(\bar{M}_w)_{ij}]_{1 \le i,j \le n_g}, \mathbf{1}_{n_g}] \in \mathbb{R}^{n_g \times (n_g+1)}$ has rank $n_g$, since the elementary matrix operations preserve the matrix rank. Since the set of all columns of $M_w$ contains all columns of $\tilde{M}_w$, this implies that $M_w$ has rank $n_g$ and $\varphi(w) \ne 0$ for this constructed particular $w$.

Therefore, the Lebesgue measure of the set $\mathcal{W} = \{w \in \mathbb{R}^d : \varphi(w) = 0\}$ is zero. If $w \notin \mathcal{W}$, $\{(f_{\bar{w},\bar{\gamma},\bar{\beta}}(g^{(1)}(x)), \ldots, f_{\bar{w},\bar{\gamma},\bar{\beta}}(g^{(n_g)}(x)) \in \mathbb{R}^{n_g} : \bar{\gamma}^{(l)}, \bar{\beta}^{(l)} \in \mathbb{R}^{ml}\} = \mathbb{R}^{n_g}$, since $M_w$ has rank $n_g$ in (1) for some $\bar{\gamma}^{(l)}, \bar{\beta}^{(l)}$ for $l = 1, \ldots, H-1$ as shown above. Thus, for any $\bar{w} \notin \mathcal{W}$ and for any $(w, \gamma, \beta)$, there exists $(\bar{\gamma}, \bar{\beta})$ such that
$$(f_{w,\gamma,\beta}(g^{(1)}(x)), \ldots, f_{w,\gamma,\beta}(g^{(n_g)}(x)) = (f_{\bar{w},\bar{\gamma},\bar{\beta}}(g^{(1)}(x)), \ldots, f_{\bar{w},\bar{\gamma},\bar{\beta}}(g^{(n_g)}(x))$$
which implies the desired statement.

$\square$

## C.2 Proof of Corollary 1

*Proof of Corollary 1.* Since non-degenerate Gaussian measure with any mean and variance is absolutely continuous with respect to Lebesgue measure, Theorem 2 implies the statement of this corollary.

$\square$

## C.3 Proof of Theorem 1

The following lemma provides an upper bound on the expected loss via expected meta-tailoring contrastive loss.

**Lemma 4.** *For every $\theta$,*
$$\mathbb{E}_{x,y}[\mathcal{L}(f_\theta(x), y)] \le \mathbb{E}_x \left[ \frac{1}{1 - \bar{\rho}(x)} \left( \mathcal{L}_{\text{cont}}^{x^+, x^-}(x, \theta^{(1:H)}) + \mathcal{L}_{\text{cont}}^{x^+, x^-, y^-}(x, \theta) - \underline{\rho}(x) \right) \right]$$

*Proof of Lemma 4.* Using the notation $\rho = \rho_y(\mu_{y^-}(x))$,

$\mathbb{E}_{x,y}[\mathcal{L}(f_\theta(x), y)]$

$= \mathbb{E}_{x,y} \left[ \frac{1}{1 - \rho} ((1 - \rho)\mathcal{L}(f_\theta(x), y) \pm \rho) \right]$

$= \mathbb{E}_{x,y} \left[ \frac{1}{1 - \rho} ((1 - \rho)\ell_{\text{cont}}(f_\theta(x)_y - f_\theta(x)_{y^- \ne y}) + \rho\ell_{\text{cont}}(f_\theta(x)_y - f_\theta(x)_{y^- = y}) - \rho) \right]$

$= \mathbb{E}_{x,y} \left[ \frac{1}{1 - \rho} \left( \mathbb{E}_{y^- \sim \mu_{y^-}(x)}[\ell_{\text{cont}}(f_\theta(x)_y - f_\theta(x)_{y^-})] - \rho \right) \right]$

$= \mathbb{E}_{x,y} \left[ \frac{1}{1 - \rho} \left( \mathbb{E}_{y^- \sim \mu_{y^-}(x)}[\psi_{x,y,y^-}(\theta^{(H+1)})] - \rho \right) \right]$

$\le \mathbb{E}_{x,y} \left[ \frac{1}{1 - \rho} \left( \psi_{x,1,2}([u_h^+, u_h^-]^\top) + \mathcal{L}_{\text{cont}}^{x^+, x^-, y^-}(x, \theta) - \rho \right) \right]$

$\le \mathbb{E}_{x,y} \left[ \frac{1}{1 - \rho} \left( \mathcal{L}_{\text{cont}}^{x^+, x^-}(x, \theta^{(1:H)}) + \mathcal{L}_{\text{cont}}^{x^+, x^-, y^-}(x, \theta) - \rho \right) \right]$

where the third line follows from the definition of $\mathcal{L}(f_\theta(x), y)$ and $\ell_{\text{cont}}(f_\theta(x)_y - f_\theta(x)_{y'=y}) = \ell_{\text{cont}}(0) = 1$, the forth line follows from the definition of $\rho$ and the expectation $\mathbb{E}_{y^- \sim \mu_{y^-}(x)}$, the fifth line follows from $f_\theta(x)_y = \theta_y^{(H+1)} h_{\theta^{(1:H)}}(x)$ and $f_\theta(x)_{y^-} = \theta_{y^-}^{(H+1)} h_{\theta^{(1:H)}}(x)$, the sixth line follows from the definition of $\mathcal{L}_{\text{cont}}^{x^+, x^-, y^-}$. The last line follows from the convexity of $\ell_{\text{cont}}$ and Jensen's inequality: i.e.,

$\psi_{x,1,2}([u_h^+, u_h^-]^\top)$

$= \ell_{\text{cont}}(\mathbb{E}_{x^+ \sim \mu_{x^+}(x)} \mathbb{E}_{x^- \sim \mu_{x^-}(x)}[(h_{\theta^{(1:H)}}(x^+) - h_{\theta^{(1:H)}}(x^-))^\top h_{\theta^{(1:H)}}(x)])$

$\le \mathbb{E}_{x^+ \sim \mu_{x^+}(x)} \mathbb{E}_{x^- \sim \mu_{x^-}(x)} \ell_{\text{cont}}((h_{\theta^{(1:H)}}(x^+) - h_{\theta^{(1:H)}}(x^-))^\top h_{\theta^{(1:H)}}(x)).$

Therefore,

$$\mathbb{E}_{x,y}[\mathcal{L}(f_\theta(x), y)]$$

$$\leq \mathbb{E}_{x,y}\left[\frac{1}{1 - \rho_y(\mu_{y^-}(x))}\left(\mathcal{L}_{\text{cont}}^{x^+, x^-}(x, \theta^{(1:H)}) + \mathcal{L}_{\text{cont}}^{x^+, x^-, y^-}(x, \theta) - \rho_y(\mu_{y^-}(x))\right)\right]$$

$$\leq \mathbb{E}_x\left[\frac{1}{1 - \bar{\rho}(x)}\left(\mathcal{L}_{\text{cont}}^{x^+, x^-}(x, \theta^{(1:H)}) + \mathcal{L}_{\text{cont}}^{x^+, x^-, y^-}(x, \theta) - \underline{\rho}(x)\right)\right]$$

where we used $\underline{\rho}(x) \leq \rho_y(\mu_{y^-}(x)) \leq \bar{\rho}(x) \in [0, 1)$. $\qquad\square$

**Lemma 5.** *Let $S \mapsto f_{\theta(x,S)}(x)$ be an uniformly $\zeta$-stable tailoring algorithm. Then, for any $\delta > 0$, with probability at least $1 - \delta$ over an i.i.d. draw of $n$ i.i.d. samples $S = ((x_i, y_i))_{i=1}^n$, the following holds:*

$$\mathbb{E}_{x,y}[\mathcal{L}(f_{\theta(x,S)}(x), y)] \leq \frac{1}{n}\sum_{i=1}^n \mathcal{L}(f_{\theta(x_i,S)}(x_i), y_i) + \frac{\zeta}{n} + (2\zeta + c)\sqrt{\frac{\ln(1/\delta)}{2n}}.$$

*Proof of Lemma 5.* Define $\varphi_1(S) = \mathbb{E}_{x,y}[\mathcal{L}(f_{\theta(x,S)}(x), y)]$ and $\varphi_2(S) = \frac{1}{n}\sum_{i=1}^n \mathcal{L}(f_{\theta(x_i,S)}(x_i), y_i)$, and $\varphi(S) = \varphi_1(S) - \varphi_2(S)$. To apply McDiarmid's inequality to $\varphi(S)$, we compute an upper bound on $|\varphi(S) - \varphi(S')|$ where $S$ and $S'$ be two training datasets differing by exactly one point of an arbitrary index $i_0$; i.e., $S_i = S_i'$ for all $i \neq i_0$ and $S_{i_0} \neq S_{i_0}'$, where $S' = ((x_i', y_i'))_{i=1}^n$. Let $\tilde{\zeta} = \frac{\zeta}{n}$ Then,

$$|\varphi(S) - \varphi(S')| \leq |\varphi_1(S) - \varphi_1(S')| + |\varphi_2(S) - \varphi_2(S')|.$$

For the first term, using the $\zeta$-stability,

$$|\varphi_1(S) - \varphi_1(S')| \leq \mathbb{E}_{x,y}[|\mathcal{L}(f_{\theta(x,S)}(x), y) - \mathcal{L}(f_{\theta(x,S')}(x), y)|]$$
$$\leq \tilde{\zeta}.$$

For the second term, using $\zeta$-stability and the upper bound $c$ on per-sample loss,

$$|\varphi_2(S) - \varphi_2(S')| \leq \frac{1}{n}\sum_{i \neq i_0}|\mathcal{L}(f_{\theta(x_i,S)}(x_i), y_i) - \mathcal{L}(f_{\theta(x_i,S')}(x_i), y_i)| + \frac{c}{n}$$

$$\leq \frac{(n-1)\tilde{\zeta}}{n} + \frac{c}{n} \leq \tilde{\zeta} + \frac{c}{n}.$$

Therefore, $|\varphi(S) - \varphi(S')| \leq 2\tilde{\zeta} + \frac{c}{n}$. By McDiarmid's inequality, for any $\delta > 0$, with probability at least $1 - \delta$,

$$\varphi(S) \leq \mathbb{E}_S[\varphi(S)] + (2\zeta + c)\sqrt{\frac{\ln(1/\delta)}{2n}}.$$

The reset of the proof bounds the first term $\mathbb{E}_S[\varphi(S)]$. By the linearity of expectation,

$$\mathbb{E}_S[\varphi(S)] = \mathbb{E}_S[\varphi_1(S)] - \mathbb{E}_S[\varphi_1(S)].$$

For the first term,

$$\mathbb{E}_S[\varphi_1(S)] = \mathbb{E}_{S,x,y}[\mathcal{L}(f_{\theta(x,S)}(x), y)].$$

For the second term, using the linearity of expectation,

$$\mathbb{E}_S[\varphi_2(S)] = \mathbb{E}_S\left[\frac{1}{n}\sum_{i=1}^n \mathcal{L}(f_{\theta(x_i,S)}(x_i), y_i)\right]$$

$$= \frac{1}{n}\sum_{i=1}^n \mathbb{E}_S[\mathcal{L}(f_{\theta(x_i,S)}(x_i), y_i)]$$

$$= \frac{1}{n}\sum_{i=1}^n \mathbb{E}_{S,x,y}[\mathcal{L}(f_{\theta(x,S_{x,y}^i)}(x), y)],$$

where $S^i$ is a sample of $n$ points such that $(S^i_{x,y})_j = S_j$ for $j \neq i$ and $(S^i_{x,y})_i = (x,y)$. By combining these, using the linearity of expectation and $\zeta$-stability,

$$\mathbb{E}_S[\varphi(S)] = \frac{1}{n}\sum_{i=1}^{n}\mathbb{E}_{S,x,y}[\mathcal{L}(f_{\theta(x,S)}(x),y) - \mathcal{L}(f_{\theta(x,S^i_{x,y})}(x),y)]$$

$$\leq \frac{1}{n}\sum_{i=1}^{n}\mathbb{E}_{S,x,y}[|\mathcal{L}(f_{\theta(x,S)}(x),y) - \mathcal{L}(f_{\theta(x,S^i_{x,y})}(x),y)|]$$

$$\leq \frac{1}{n}\sum_{i=1}^{n}\tilde{\zeta} = \tilde{\zeta}.$$

Therefore, $\mathbb{E}_S[\varphi(S)] \leq \tilde{\zeta}$.

$\square$

*Proof of Theorem 1.* For any $\theta$ and $\kappa \in [0,1]$,

$$\mathbb{E}_{x,y}[\mathcal{L}(f_\theta(x),y)] = \kappa\mathbb{E}_{x,y}[\mathcal{L}(f_\theta(x),y)] + (1-\kappa)\mathbb{E}_{x,y}[\mathcal{L}(f_\theta(x),y)].$$

Applying Lemma 4 for the first term and Lemma 5 yields the desired statement.

$\square$

### C.4  Statement and proof of Theorem 6

**Theorem 6.** *Let $\mathcal{F}$ be an arbitrary set of maps $x \mapsto f_{\theta_x}(x)$. Then, for any $\delta > 0$, with probability at least $1 - \delta$ over an i.i.d. draw of $n$ i.i.d. samples $((x_i, y_i))_{i=1}^n$, the following holds: for all maps $(x \mapsto f_{\theta_x}(x)) \in \mathcal{F}$ and any $\kappa \in [0,1]$, we have that $\mathbb{E}_{x,y}[\mathcal{L}^{\mathrm{sup}}(f_{\theta_x}(x),y)] \leq \kappa\mathbb{E}_x[\mathcal{L}_{\mathrm{cont}}(x,\theta_x)] + (1-\kappa)\mathcal{J}'$, where $\mathcal{J}' = \frac{1}{n}\sum_{i=1}^n \mathcal{L}^{\mathrm{sup}}(f_{\theta_{x_i}}(x_i), y_i) + 2\mathcal{R}_n(\mathcal{L}^{\mathrm{sup}} \circ \mathcal{F}) + c\sqrt{(\ln(1/\delta))/(2n)}$.*

The following lemma is used along with Lemma 4 to prove the statement of this theorem.

**Lemma 7.** *Let $\mathcal{F}$ be an arbitrary set of maps $x \mapsto f_{\theta_x}(x)$. For any $\delta > 0$, with probability at least $1 - \delta$ over an i.i.d. draw of $n$ i.i.d. samples $((x_i, y_i))_{i=1}^n$, the following holds: for all maps $(x \mapsto f_{\theta_x}(x)) \in \mathcal{F}$,*

$$\mathbb{E}_{x,y}[\mathcal{L}(f_{\theta_x}(x),y)] \leq \frac{1}{n}\sum_{i=1}^{n}\mathcal{L}(f_{\theta_{x_i}}(x_i), y_i) + 2\mathcal{R}_n(\mathcal{L} \circ \mathcal{F}) + c\sqrt{\frac{\ln(1/\delta)}{2n}}.$$

*Proof of Lemma 7.* Let $S = ((x_i, y_i))_{i=1}^n$ and $S' = ((x'_i, y'_i))_{i=1}^n$. Define

$$\varphi(S) = \sup_{(x \mapsto f_{\theta_x}(x)) \in \mathcal{F}} \mathbb{E}_{x,y}[\mathcal{L}(f_{\theta_x}(x),y)] - \frac{1}{n}\sum_{i=1}^{n}\mathcal{L}(f_{\theta_{x_i}}(x_i), y_i).$$

To apply McDiarmid's inequality to $\varphi(S)$, we compute an upper bound on $|\varphi(S) - \varphi(S')|$ where $S$ and $S'$ be two training datasets differing by exactly one point of an arbitrary index $i_0$; i.e., $S_i = S'_i$ for all $i \neq i_0$ and $S_{i_0} \neq S'_{i_0}$. Then,

$$\varphi(S') - \varphi(S) \leq \sup_{(x \mapsto f_{\theta_x}(x)) \in \mathcal{F}} \frac{\mathcal{L}(f_{\theta(x_{i_0})}(x_{i_0}), y_{i_0}) - \mathcal{L}(f_{\theta(x'_{i_0})}(x'_{i_0}), y'_{i_0})}{n} \leq \frac{c}{n}.$$

Similarly, $\varphi(S) - \varphi(S') \leq \frac{c}{n}$. Thus, by McDiarmid's inequality, for any $\delta > 0$, with probability at least $1 - \delta$,

$$\varphi(S) \leq \mathbb{E}_S[\varphi(S)] + c\sqrt{\frac{\ln(1/\delta)}{2n}}.$$

Moreover, with $f(x) = f_{\theta_x}(x)$,

$$\mathbb{E}_S[\varphi(S)] = \mathbb{E}_S\left[\sup_{f \in \mathcal{F}} \mathbb{E}_{S'}\left[\frac{1}{n}\sum_{i=1}^{n}\mathcal{L}(f_{\theta(x_i')}(x_i'), y_i') \right] - \frac{1}{n}\sum_{i=1}^{n}\mathcal{L}(f_{\theta_{x_i}}(x_i), y_i)\right]$$

$$\leq \mathbb{E}_{S,S'}\left[\sup_{f \in \mathcal{F}} \frac{1}{n}\sum_{i=1}^{n}(\mathcal{L}(f_{\theta(x_i')}(x_i'), y_i') - \mathcal{L}(f_{\theta_{x_i}}(x_i), y_i)\right]$$

$$\leq \mathbb{E}_{\xi,S,S'}\left[\sup_{f \in \mathcal{F}} \frac{1}{n}\sum_{i=1}^{n}\xi_i(\mathcal{L}(f_{\theta(x_i')}(x_i'), y_i') - \mathcal{L}(f_{\theta_{x_i}}(x_i), y_i))\right]$$

$$\leq 2\mathbb{E}_{\xi,S}\left[\sup_{f \in \mathcal{F}} \frac{1}{n}\sum_{i=1}^{n}\xi_i\mathcal{L}(f_{\theta_{x_i}}(x_i), y_i))\right]$$

where the fist line follows the definitions of each term, the second line uses the Jensen's inequality and the convexity of the supremum, and the third line follows that for each $\xi_i \in \{-1, +1\}$, the distribution of each term $\xi_i(\mathcal{L}(f_{\theta(x_i')}(x_i'), y_i') - \mathcal{L}(f_{\theta_{x_i}}(x_i), y_i))$ is the distribution of $(\mathcal{L}(f_{\theta(x_i')}(x_i'), y_i') - \mathcal{L}(f_{\theta_{x_i}}(x_i), y_i))$ since $\bar{S}$ and $\bar{S}'$ are drawn iid with the same distribution. The forth line uses the subadditivity of supremum. $\qquad\square$

*Proof of Theorem 6.* For any $\theta$ and $\kappa \in [0, 1]$,
$$\mathbb{E}_{x,y}[\mathcal{L}(f_\theta(x), y)] = \kappa\mathbb{E}_{x,y}[\mathcal{L}(f_\theta(x), y)] + (1 - \kappa)\mathbb{E}_{x,y}[\mathcal{L}(f_\theta(x), y)].$$
Applying Lemma 4 for the first term and Lemma 7 yields the desired statement.

$\qquad\square$

## C.5  Proof of Theorem 3

*Proof of Theorem 3.* Let $\theta^{(1:H)}$ be fixed. We first prove the first statement for the convexity. The function $\theta^{(H+1)} \mapsto \psi_{x,y,y^-}(\theta^{(H+1)})$ is convex, since it is a composition of a convex function $\ell_{\text{cont}}$ and a affine function $z \mapsto (z_y - z_{y^-})h_{\theta(1:H)}(x)$. The function $\theta^{(H+1)} \mapsto \mathbb{E}_{y^- \sim \mu_{y^-}(x)}[\psi_{x,y,y^-}(\theta^{(H+1)}) - \psi_{x,1,2}([u_h^+, u_h^-]^\top)]$ is convex since the expectation and affine translation preserves the convexity. Finally, $\theta^{(H+1)} \mapsto \mathcal{L}_{\text{cont}}^{x^+,x^-,y^-}(x, \theta^{(1:H)}, \theta^{(H+1)})$ is convex since it is the piecewise maximum of the convex functions
$$\theta^{(H+1)} \mapsto \mathbb{E}_{y^- \sim \mu_{y^-}(x)}[\psi_{x,y,y^-}(\theta^{(H+1)}) - \psi_{x,1,2}([u_h^+, u_h^-]^\top)]$$
for each $y$.

We now prove the second statement of the theorem for the inequality. Let us write $\mu_{x^+} = \mu_{x|y}$ and $\mu_{x^-} = \mu_{x|y^-}$. Let $U = [u_1, u_2]^\top \in \mathbb{R}^{m_y \times (m_H+1)}$ where $u_y = \mathbb{E}_{x \sim \mu_{x|y}}[h_{\theta(1:H)}(x)]$ for $y \in \{1, 2\}$. Then,
$$u_h^+ = \mathbb{E}_{x^+ \sim \mu_{x^+}(x)}[h_{\theta(1:H)}(x^+)] = \mathbb{E}_{x^+ \sim \mu_{x|y}}[h_{\theta(1:H)}(x^+)] = u_y,$$
and
$$u_h^- = \mathbb{E}_{x^- \sim \mu_{x^-}(x)}[h_{\theta(1:H)}(x^-)] = \mathbb{E}_{x^- \sim \mu_{x|y^-}}[h_{\theta(1:H)}(x^-)] = u_{y^-}.$$
Therefore,
$$\psi_{x,1,2}([u_h^+, u_h^-]^\top) = \psi_{x,y,y^-}(U),$$
with which
$$\mathcal{L}_{\text{cont}}^{x^+,x^-,y^-}(x, \theta) = \max_y \mathbb{E}_{y^- \sim \mu_{y^-}(x)}[\psi_{x,y,y^-}(\theta^{(H+1)}) - \psi_{x,y,y^-}(U)].$$
Since $U$ and $\theta^{(1:H)}$ do not contain $\theta^{(H+1)}$, for any $U, \bar{\theta}^{(1:H)}$, there exists $\theta^{(H+1)} = U$ for which $\psi_{x,y,y^-}(\theta^{(H+1)}) - \psi_{x,y,y^-}(U) = 0$ and hence $\mathcal{L}_{\text{cont}}^{x^+,x^-,y^-}(x, \theta) = 0$. Therefore,
$$\inf_{\theta^{(H+1)} \in \mathbb{R}^{m_y \times (m_H+1)}} \mathcal{L}_{\text{cont}}^{x^+,x^-}(x, \theta^{(1:H)}) + \mathcal{L}_{\text{cont}}^{x^+,x^-,y^-}(x, \theta^{(1:H)}, \theta^{(H+1)})$$
$$= \mathcal{L}_{\text{cont}}^{x^+,x^-}(x, \theta^{(1:H)}) + \inf_{\theta^{(H+1)} \in \mathbb{R}^{m_y \times (m_H+1)}} \mathcal{L}_{\text{cont}}^{x^+,x^-,y^-}(x, \theta^{(1:H)}, \theta^{(H+1)})$$
$$\leq \mathcal{L}_{\text{cont}}^{x^+,x^-}(x, \theta^{(1:H)}).$$

$\qquad\square$

## C.6  Proof of Remark 1

*Proof of Remark 1.* For any $\theta$,

$$\mathbb{E}_{x,y}[\mathcal{L}(f_\theta(x), y)] = \inf_{\kappa \in [0,1]} \kappa \mathbb{E}_{x,y}[\mathcal{L}(f_\theta(x), y)] + (1 - \kappa)\mathbb{E}_{x,y}[\mathcal{L}(f_\theta(x), y)].$$

Applying Lemma 5 for Theorem 1 (and Lemma 7 for Theorem 6) to the second term and the assumption $\mathbb{E}_{x,y}[\mathcal{L}(f_\theta(x), y)] \leq \mathbb{E}_x[\mathcal{L}_{\text{un}}(f_\theta(x))]$ to the first term yields the desired statement.

$\square$

# D   Details and description of CNGRAD

In this section we describe CNGRAD in greater detail: its implementation, different variants and run-time costs. Note that, although this section is written from the perspective of meta-tailoring, CN-GRAD is also applicable to meta-learning, we provide pseudo-code in algorithm 4. The main idea behind CNGRAD is to optimize only conditional normalization (CN) parameters $\gamma^{(l)}, \beta^{(l)}$ in the inner loop and optimize all the other weights $w$ in the outer loop. To simplify notation for implementation, in this subsection only, we overload notations to make them work over a mini-batch as follows. Let $b$ be a (mini-)batch size. Given $X \in \mathbb{R}^{b \times m_0}$, $\gamma \in \mathbb{R}^{b \times \sum_l m_l}$ and $\beta \in \mathbb{R}^{b \times \sum_l m_l}$, let $(f_{w,\gamma,\beta}(X))_i = f_{w,\gamma_i,\beta_i}(X_i)$ where $X_i$, $\gamma_i$, and $\beta_i$ are the transposes of the $i$-th row vectors of $X$, $\gamma$ and $\beta$, respectively. Similarly, $\mathcal{L}^{\text{sup}}$ and $\mathcal{L}^{\text{tailor}}$ are used over a mini-batch. We also refer to $\theta = (w, \gamma, \beta)$.

**Initialization of** $\gamma, \beta$   In the inner loop we always initialize $\gamma = \mathbf{1}_{b,\sum_l m_l}, \beta = \mathbf{0}_{b,\sum_l m_l}$. More complex methods where the initialization of these parameters is meta-trained are also possible. However, we note two things:

1. By initializing to the identity function, we can pick an architecture trained with regular inductive learning, add CN layers without changing predictions and perform tailoring. In this manner, the prediction algorithm is the same regardless of whether we trained with meta-tailoring or without the CN parameters.

2. We can add a previous normalization layer with weights $\gamma'^{(l)}, \beta'^{(l)}$ that are trained in the outer loop, having a similar effect than meta-learning an initialization. However, we do not do it in our experiments.

**First and second order versions of** CNGRAD:   $w$ affect $\mathcal{L}^{\text{sup}}$ in two ways: first, they directly affect the evaluation $f_{w,\gamma_s,\beta_s}(X)$ by being weights of the neural network; second, they affect $\nabla_\beta \mathcal{L}^{\text{tailor}}, \nabla_\gamma \mathcal{L}^{\text{tailor}}$ which affects $\gamma_s, \beta_s$ which in turn affect $\mathcal{L}^{\text{sup}}$. Similar to MAML [19], we can implement two versions: in the first order version we only take into account the first effect, while in the second order version we take into account both effects. The first order version has three advantages:

1. It is very easy to code: the optimization of the inner parameters and the outer parameters are detached and we simply need to back-propagate $\mathcal{L}^{\text{tailor}}$ with respect to $\beta, \gamma$ and $\mathcal{L}^{\text{sup}}$ with respect to $w$. This version is easier to implement than most meta-learning algorithms, since the parameters in the inner and outer loop are different.

2. It is faster: because we do not back-propagate through the optimization, the overall computation graph is smaller.

3. It is more stable to train: second-order gradients can be a bit unstable to train; this required us to lower the inner tailoring learning rate in experiments of section 5.1 for the second-order version.

The second-order version has one big advantage: it optimizes the true objective, taking into account how $\mathcal{L}^{\text{tailor}}$ will affect the update of the network. This is critical to linking the unsupervised loss to best serve the supervised loss by performing informative updates to the CN parameters.

**WarpGrad-inspired stopping of gradients and subsequent reduction in memory cost:**   Warp-Grad [20] was an inspiration to CNGRAD suggesting to interleave layers that are adapted in the inner loop with layers only adapted in the outer loop. In contrast to WarpGrad, we can evaluate inputs (in meta-tailoring) or tasks (in meta-learning) in parallel, which speeds up training and inference. This also simplifies the code because we do not have to manually perform batches of tasks by iterating through them.

WarpGrad also proposes to stop the gradients between inner steps; we include this idea as an optional operation in CNGRAD, as shown in line 12 of 3. The advantage of adding it is that it decreases the memory cost when performing multiple inner steps, as we now only have to keep in memory the computation graph of the last step instead of all the steps, key when the networks are very deep like in the experiments of section 5.4. Another advantage is that it makes training more stable, reducing

**Algorithm 3** CNGRAD for meta-tailoring

---

**Subroutine** *Training($f$, $\mathcal{L}^{\text{sup}}$, $\lambda_{sup}$, $\mathcal{L}^{\text{tailor}}$, $\lambda_{tailor}$, steps,$((x_i, y_i))_{i=1}^n$)*

 randomly initialize $w$    `// All parameters except` $\gamma, \beta$`; trained in outer loop`
 **while** *not done* **do**
  **for** $0 \leq i \leq n/b$ **do**                 `// b batch size`
   $X, Y = x_{ib:i(b+1)}, y_{ib:i(b+1)}$   $\gamma_0 = \mathbf{1}_{b, \sum_l m_l}$   $\beta_0 = \mathbf{0}_{b, \sum_l m_l}$
   **for** $1 \leq s \leq steps$ **do**
    $\gamma_s = \gamma_{s-1} - \lambda_{tailor} \nabla_\gamma \mathcal{L}^{\text{tailor}}(w, \gamma_{s-1}, \beta_{s-1}, X)$   `// Inner step w.r.t.` $\gamma$
    $\beta_s = \beta_{s-1} - \lambda_{tailor} \nabla_\beta \mathcal{L}^{\text{tailor}}(w, \gamma_{s-1}, \beta_{s-1}, X)$   `// Inner step w.r.t.` $\beta$
    $\gamma_s, \beta_s = \gamma_s.detach(), \beta_s.detach()$   `// Optional operation, only in` $1^{st}$
    `order CNGrad:  WarpGrad detach to avoid back-proping through`
    `multiple steps; reducing memory, and increasing stability, but`
    `adding bias.`
    $w = w - \lambda_{sup} \nabla_w \mathcal{L}^{\text{sup}}\left(f_{w, \gamma_s, \beta_s}(X), Y)\right)$        `// Outer step`
 **return** $w$

**Subroutine** *Prediction($f$, $w$, $\mathcal{L}^{\text{tailor}}$, $\lambda$, steps, $X$)*   `// For meta-tailoring & tailoring`
     `// X contains multiple inputs, with independent tailoring processes`
 $b = X.shape[0]$                   `// number of inputs`
 $\gamma_0 = \mathbf{1}_{b, \sum_l m_l}$   $\beta_0 = \mathbf{0}_{b, \sum_l m_l}$
 **for** $1 \leq s \leq steps$ **do**
  $\gamma_s = \gamma_{s-1} - \lambda \nabla_\gamma \mathcal{L}^{\text{tailor}}(w, \gamma_{s-1}, \beta_{s-1}, X)$
  $\beta_s = \beta_{s-1} - \lambda \nabla_\beta \mathcal{L}^{\text{tailor}}(w, \gamma_{s-1}, \beta_{s-1}, X)$
 **return** $f_{w, \gamma_{steps}, \beta_{steps}}(X)$

---

variance, as back-propagating through the optimization is often very noisy for many steps. At the same time it adds bias, because it makes the greedy assumption that locally minimizing the decrease in outer loss at every step will lead to low overall loss after multiple steps.

**Computational cost:** in CNGRAD we perform multiple forward and backward passes, compared to a single forward pass in the usual setting. In particular, if we perform $s$ tailoring steps, we execute $(s + 1)$ forward steps and $s$ backward steps, which usually take the same amount of time as the forward steps. Therefore, in its naive implementation, this method takes about $2s + 1$ times more than executing the regular network without tailoring.

However, it is well-known that we can often only adapt the higher layers of a network, while keeping the lower layers constant. Moreover, our proof about the capacity of CNGRAD to optimize a broad range of inner losses only required us to adapt the very last CN layer $\gamma^{(H)}, \beta^{(H)}$. This implies we can put the CN layers only on the top layer(s). In the case of only having one CN layer at the last network layer, we only require one initial full forward pass (as we do without tailoring). Then, we have $s$ backward-forward steps that affect only the last layer, thus costing $\frac{1}{H}$ in case of layers of equivalent cost. This leads to a factor of $1 + \frac{2s}{H}$ in cost, which for $s$ small and $H$ large (typical for deep networks), is a very small overcost. Moreover, for tailoring and meta-tailoring, we are likely to get the same performance with smaller networks, which may compensate the increase in cost.

**Meta-learning version:** CNGRAD can also be used in meta-learning, with the advantage of being provably expressive, very efficient in terms of parameters and compute, and being able to parallelize across tasks. We show the pseudo-code for few-shot supervised learning in algorithm 4. There are two changes to handle the meta-learning setting: first, in the inner loop, instead of the unsupervised tailoring loss we optimize a supervised loss on the training (support) set. Second, we want to share the same inner parameters $\gamma, \beta$ for different samples of the same task. To do so we add the operation "repeat_interleave" (PyTorch notation), which makes $k$ contiguous copies of each parameter $\gamma, \beta$, before feeding them to the network evaluation. In doing so, gradients coming from different samples of the same task get pooled together. At test time we do the same for the $k'$ queries ($k'$ can be different than $k$). Note that, in practice, this pooling is also used in meta-tailoring when we have more than one data augmentation within $\mathcal{L}^{\text{tailor}}$.

**Algorithm 4** CNGRAD for meta-learning

**Subroutine** *Meta-training(f, $\mathcal{L}^{\text{sup}}$, $\lambda_{inner}$, $\lambda_{outer}$, steps,$\mathcal{T}$)*

    randomly initialize $w$      // All parameters except $\gamma, \beta$; trained in outer loop

    **while** *not done* **do**

        **for** $0 \leq i \leq n/b$ **do**                               // $b$ batch size

            $X_{train}, Y_{train} = [\,], [\,]$   $X_{test}, Y_{test} = [\,], [\,]$ **for** $ib \leq j \leq i(b+1)$ **do**

                $(inp, out) \sim_k \mathcal{T}_j$      // Take $k$ samples from each task for training

                $X.append\,(inp)\,; Y.append\,(out)$   $(query, target) \sim'_k \mathcal{T}_j$   // Take $k'$ samples

                from each task for testing

                $X.append\,(query)\,; Y.append\,(target)$

                            // We can now batch evaluations of multiple tasks

            $X_{train}, Y_{train} = concat\,(X_{train}, dim = 0)\,, concat\,(Y_{train}, dim = 0)$

            $X_{test}, Y_{test} = concat\,(X_{test}, dim = 0)\,, concat\,(Y_{test}, dim = 0)$   $\gamma_0 = \mathbf{1}_{b, \sum_l m_l}$

            $\beta_0 = \mathbf{0}_{b, \sum_l m_l}$ **for** $1 \leq s \leq steps$ **do**

                // We now repeat the CN parameters $k$ times so that samples from
                the same task share the same CN parameters

                $\gamma^{tr}_{s-1}, \beta^{tr}_{s-1} = \gamma_{s-1}.repeat\_interleave(k, 1), \beta_{s-1}.repeat\_interleave(k, 1)$

                $\gamma_s = \gamma_{s-1} - \lambda_{innner}\nabla_\gamma \mathcal{L}^{\text{sup}}(f_{w, \gamma^{tr}_{s-1}, \beta^{tr}_{s-1}}(X_{train}), Y_{train})$

                $\beta_s = \beta_{s-1} - \lambda_{innner}\nabla_\beta \mathcal{L}^{\text{sup}}(f_{w, \gamma^{tr}_{s-1}, \beta^{tr}_{s-1}}(X_{train}), Y_{train})$

                $\gamma^{test}_s, \beta^{test}_s = \gamma_s.repeat\_interleave(k', 1), \beta_s.repeat\_interleave(k', 1)$

                $w = w - \lambda_{outer}\nabla_w \mathcal{L}^{\text{sup}}\left(f_{w, \gamma^{test}_s, \beta^{test}_s}(X_{test}), Y_{test})\right)$

                $\beta_s, \gamma_s = \beta_s.detach(), \gamma_s.detach()$     // WarpGrad detach to not backprop
                through multiple steps

    **return** $w$

**Subroutine** *Meta-test(f, w, $\mathcal{L}^{\text{sup}}$, $\lambda_{inner}$,steps,$X_{train}$, $Y_{train}$, $X_{test}$)*

    // Assuming a single task, although we could evaluate multiple tasks in
    parallel as in meta-training.

    $\gamma_0 = \mathbf{1}_{1, \sum_l m_l}$                       // single $\gamma, \beta$ because we only have one task

    $\beta_0 = \mathbf{0}_{1, \sum_l m_l}$ **for** $1 \leq s \leq steps$ **do**

        $\gamma^{tr}_{s-1}, \beta^{tr}_{s-1} = \gamma_{s-1}.repeat\_interleave(k, 1), \beta_{s-1}.repeat\_interleave(k, 1)$

        $\gamma_s = \gamma_{s-1} - \lambda_{innner}\nabla_\gamma \mathcal{L}^{\text{sup}}(f_{w, \gamma^{tr}_{s-1}, \beta^{tr}_{s-1}}(X_{train}), Y_{train})$

        $\beta_s = \beta_{s-1} - \lambda_{innner}\nabla_\beta \mathcal{L}^{\text{sup}}(f_{w, \gamma^{tr}_{s-1}, \beta^{tr}_{s-1}}(X_{train}), Y_{train})$

    $\gamma^{test}_{steps}, \beta^{test}_{steps} = \gamma_{steps}.repeat\_interleave(k', 1), \beta_{steps}.repeat\_interleave(k', 1)$ **return**
    $f_{w, \gamma^{test}_{steps}, \beta^{test}_{steps}}(X_{test})$

# E   Experimental details of physics experiments

**Dataset generation**   As mentioned in the main text, 5-body systems are chaotic and most random configurations are unstable. To generate our dataset we used Finite Differences to optimize 5-body dynamical systems that were stable for 200 steps (no planet collisions and no planet outside a predetermined grid) and then picked the first 100 steps of their trajectories, to ensure dynamical stability. To generate each trajectory, we randomly initialized 5 planets within a 2D grid of size $w = 600, h = 300$, with a uniform probability of being anywhere in the central grid of size $w/2, h/2$, each with a mass sampled from a uniform between $[0.15, 0.25]$ (arbitrary units) and with random starting velocity initialized with a Gaussian distribution. We then use a 4th order Runge-Kutta integrator to accurately simulate the ODE of the dynamical system until we either reach 200 steps, two planets get within a certain critical distance from each other or a planet gets outside the pre-configured grid. If the trajectory reached 200 steps, we added it to the dataset; otherwise we made a small random perturbation to the initial configuration of the planets and tried again. If the new perturbation did not reach 200 steps, but lasted longer we kept the perturbation as the new origin for future initialization perturbations, otherwise we kept our current initialization. Once all the datasets were generated we picked those below a threshold mean mass and partitioned them randomly into train and test. Finally, we normalize each of the 25 dimensions (5 planets and for each planet $x,y,v_x,v_y,m$) to have mean zero and standard deviation one. For inputs, we use each state and as target we use the next state; therefore, each trajectory gives us 100 pairs.

For more details, we attach the code that generated the dataset.

**Implementation of tailoring, meta-tailoring and** CNGRAD   All of our code is implemented in PyTorch [38], using the higher library [22](`https://github.com/facebookresearch/higher`) to implement the second-order version of CNGRAD. We implemented a 3-layer feedforward neural network, with a conditional normalization layer after each layer except the final regression layer. The result of the network was added to the input, thus effectively predicting the delta between the current state and the next state. For both the first-order and second-order versions of CNGRAD, we used the detachment of WarpGrad (line 12 in algorithm 3). For more details, we also attach the implementation of the method.

**Compute and hyper-parameter search**   To keep the evaluation as strict as possible, we searched all the hyper-parameters affecting the inductive baseline and our tailoring versions with the baseline and simply copied these values for tailoring and meta-tailoring. For the latter two, we also had to search for $\lambda_{tailor}$.

The number of epochs was 1000, selected with the inductive baseline, although more epochs did not substantially affect performance in either direction. We note that meta-tailoring performance plateaued earlier in terms of epochs, but we left it the same for consistency. Interestingly, we found that regularizing the physics loss (energy and momentum conservation) helped the inductive baseline, even though the training data already has 0 physics loss. We searched over $[10^{-4}, 3 \cdot 10^{-4}, 10^{-3}, 3 \cdot 10^{-3}, 10^{-2}]$ for the weight assigned to the physics loss and chose $2 \cdot 10^{-3}$ for best performance in the inductive baseline. To balance between energy and momentum losses we multiplied the momentum loss by 10 to roughly balance their magnitudes before adding them into a single physics loss, this weighting was not searched. We copied these settings for meta-tailoring.

In terms of the neural network architecture, we chose a simple model with 3 hidden layers of the same size and tried $[128, 256, 512]$ on the inductive baseline, choosing 512 and deciding not to go higher for compute reasons and because we were already able to get much lower training loss than test loss. We copied these settings for the meta-tailoring setup. We note that since there are approximately $O(m_h^2)$ weight parameters, yet only $O(m_h)$ affine parameters used for tailoring, adding tailoring and meta-tailoring increase parameters roughly by a fraction $O(1/m_h)$, or about $0.2\%$. Also in the inductive baseline, we tried adding Batch Normalization [30], since it didn't affect performance we decided not to add it.

We chose the tailoring step size parameter by trying $[10^{-5}, 10^{-4}, 10^{-3}, 10^{-2}]$, finding $10^{-3}$ worked well while requiring less steps than using a smaller step size. We therefore used this step for meta-tailoring as well, which worked well for first-order CNGRAD, but not for second-order CNGRAD, whose training diverged. We thus lowered the tailoring step size to $10^{-4}$ for the second-order version, which worked well. We also tried clipping the inner gradients to increase the stability of the second-

order method; since gains on preliminary experiments were small, we decided to keep it out for simplicity.

For meta-tailoring we only tried 2 and 5 tailoring steps (we wanted more than one step to show the algorithm capability, but few tailoring steps to keep inference time competitive). Since they worked similarly well, we chose 2 steps to have a faster model. For the second-order version we also used 2 steps, which performed much better than the inductive baseline and tailoring, but worse than the first-order version reported in the main text(about $20\%$ improvement of the second-order version vs. $7\%$ of tailoring and $35\%$ improvement of the first-order version).

For the baseline of optimizing the output we tried a step size of $10^{-4}, 10^{-3}, 10^{-2}, 10^{-1}$. Except for a step size of $10^{-1}$, results optimized the physics loss and always achieved a very small improvement, without overfitting to the physics loss. We thus chose a big learning rate and high number of steps to report the biggest improvement, of $0.7\%$.

**Runs, compute and statistical confidence:**  we ran each method 2 times and averaged the results. Note that the baseline of optimizing the output and *tailoring* start from the inductive learning baseline, as they are meant to be methods executed after regular inductive training. This is why both curves start at the same point in Figure 2. For those methods, we report the standard deviation of the mean estimate of the *improvement*, since they are executed on the same run. Note that the standard deviation of the runs would be higher, but less appropriate. For meta-tailoring, we do use the standard deviation of the mean estimate of both runs, since they are independent from the inductive baseline.

All experiments were performed on a GTX 2080 Ti with 4 CPU cores.

## F   Experimental details on real pendulum

We modify the real pendulum of Hamiltonian Neural Networks [23]. In particular we pick the energy of the system and use it as a tailoring loss. Greydanus et al. [23] train a vanilla MLP and show that its non-conservation of energy results in poor generalization from train to test for long-term predictions. With HNNs that automatically discover an energy function and encode hamiltonian dynamics into the network architecture, the system conserves this proxy energy even in long predictions, resulting in better generalization. We meta-tailor the vanilla MLP, with no change in its architecture, beyond adding CN layers to efficiently perform tailoring. We try different inner learning rates ($1e-3, 1e-2, 1e-1, 1e0$) as well as number of steps ($1, 2, 3$) and evaluate on long-term *training* loss. Since training is 4 times as long as test, we divide training into 4 equally-big trajectories and choose the configuration with the best loss: 3 steps and $1e-1$ inner learning rate. It is worth noting that these long term predictions use scipy's ODE integrator, which is also used for the vanilla MLP as well as for HNNs. We see that, by not fully enforcing energy conservation, meta-tailoring improves over both an inductive baseline of the same architecture and HNNs.

Experiments were performed with a Volta V-100 and 10 CPU cores, taking a couple of hours to run in total.

## G   Experimental details on contrastive learning

We take the implementation of SimCLR [13] from `https://github.com/leftthomas/SimCLR` evaluating on CIFAR-10 [31].

As detailed in the main text we train the vanilla SimCLR to get an unsupervised representation. We than train *only* the linear layer with different amounts of training data, from 50 to 5000 points per class. Vanilla SimCLR follows regular inductive learning with supervised labels for the linear layer. Meta-tailoring uses the same augmentations provided by SimCLR and minimizes the SimCLR loss on each particular input before feeding the tailored representations to the linear layer. The linear layer is trained to take these adapted representations. We tried different hyper-parameters: $[4, 8, 16]$ augmentations, $1, 2,$ inner optimization steps, inner learning rate of $[1e-1, 3e-1, 1e0, 3e0, 1e1, 3e1]$ and whether to tailor the CN layers of the CNN representation or tailor the representations $h$ directly. We found very consistent results where all stable inner optimizations improved over vanilla SimCLR, and longer optimizations with larger learning rates and more augmentations gave bigger improvements. Tailoring the CN layers or the representations directly didn't make a big effect, the latter being slightly more

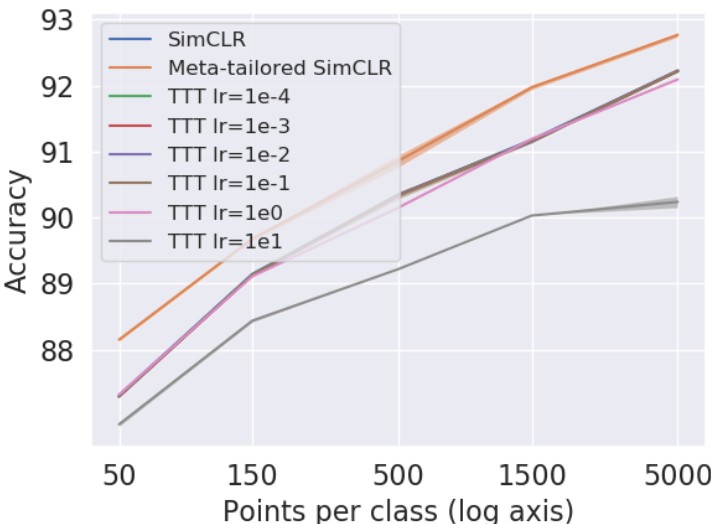

Figure 5: Test-time training (TTT) with its original rotation-prediction auxiliary task performs worse than vanilla SimCLR. Performance degrades as we increase the inner learning rate, thus increasing its power.

stable, and providing somewhat larger results. It is also much faster as we do not need to back-prop back through the CNN. We thus chose 3 steps $1e1$ learning rate, 16 augmentations and tailoring the representations directly. For TTT we kept the 3 steps and tried $0, 1e-4, 1e-3, 1e-2, 1e-1, 1e0, 1e1$ learning rates. We noticed that all these inner optimizations were stable, yet performance degraded with learning rate. Thus the best learning rate was $0$, equivalent to not doing TTT.

For all methods we follow the code-base and keep the best validation. Keeping running averages or choosing concrete epochs gave very similar results because of the stability of training a linear layer. We averaged over 5 different trainings of the linear layer, all using the same SimCLR base. The TTT baseline uses the best SimCLR epoch for each of these 5 runs, so that using TTT with lr=0 gives exactly the same results as the baseline.

For TTT we trained on the rotation prediction task proposed in the original paper. To minimize differences, we use the same architecture as the MLP from SimCLR, except with 4 output logits, one per rotation. It achieves $80.5\%$ test accuracy on rotation prediction. TTT proposes to back-propagate this rotation prediction loss back at test-time, but does not take this procedure into account at training time. We consistently find that TTT worsens the performance, with the gap becoming worse as we increase the learning rate. We think the reason why this loss is helpful in a very similar dataset and architecture in Sun et al. [46], yet hurts performance in this case is due to two factors. First, it can be observed in the original paper that rotation-prediction provides consistent, but small gains, in the 1-sample case, with much larger gains in its online multi-sample version. Second, Sun et al. [46] focus on out-of-distribution generalization, where weights are trained on a different data distribution and are thus sub-optimal. The linear layer receiving OOD inputs in this same-distribution case hurts performance, but in their OOD application, even the unmodified inputs were already OOD. Meta-tailoring takes the adaptation into account, thus the inputs of the linear layer are always in-distribution, thus being able to help performance.

Experiments were performed with a set of Volta V-100 and 10 CPU cores. SimCLR takes around a day to train. All other experiments training the linear layer from different initializations and for all data quantities take a few hours for a single set of hyper-parameters.

## H   Toy adversarial examples experiment

We illustrate the tailoring process with a simple illuminating example using the data from Ilyas et al. [29]. They use discriminant analysis on the data in Figure 6 and obtain the purple linear separator. It

has the property that, under assumptions about Gaussian distribution of the data, points above the line are more likely to have come from the blue class, and those below, from the red class. This separator is not very adversarially robust, in the sense that, for many points, a perturbation with a small $\delta$ would change the assigned class. We improve the robustness of this classifier by tailoring it using the loss

$\mathcal{L}^{\text{tailor}}(x, \theta) = \text{KL}(\phi(f_\theta(x)) \parallel \phi(f_\theta(x + \text{argmax}_{|\delta| < \varepsilon} \sum_j e^{f_\theta(x + \delta)_j})))$, where KL represents the KL divergence, $\phi$ is the logistic function, and $\phi(f_\theta(x)_i)$ is the probability of $x$ being in class $i$, so that $\phi(f_\theta(x))$ represents the entire class distribution.

With this loss, we can adjust our parameters $\theta$ so that the KL divergence between our prediction at $x$ is closer to the prediction at perturbed point $x + \delta$, over all perturbations in radius $\varepsilon$. Note that we initialized the models with the weights of Cohen et al. [14] to speed up training in all ImageNet experiments and to avoid training divergence for CIFAR-10 with $\sigma = 1$ (this divergence was already noted by Zhai et al. [53]). Each of the curves in Figure 3 corresponds to a decision boundary induced by tailoring the original separator with a different value for the maximum perturbation $\varepsilon$. Note that the resulting separators are non-linear, even though we are tailoring a linear separator, because the tailoring is local to the prediction point. We also have the advantage of being able to choose different values of $\varepsilon$ at prediction time.

**Hyper-parameters** the model does not have any hyper-parameters, as we use the model from Ilyas et al. [29], which is based on the mean $\mu$ and standard deviation $\sigma$ of the Gaussians. For tailoring, we used a $5 \times 5$ grid to initialize the inner optimization to find the point of highest probability within the $\epsilon$-ball. Using a single starting point did not work as gradient descent found a local optima. Using more ($10 \times 10$) did not improve results further, while increasing compute. We also experimented between doing a weighted average of the predictions by their energy to compute the tailoring loss or picking the element with the biggest energy. Results did not seem to differ much (likely because likelihood distributions are very peaked), so we picked the simplest option of imitating the element of highest probability. Doing a single tailoring step already worked well (we tried step sizes of $10^{-1}, 1, 10, 30$ with 10 working best), so we kept that for simplicity and faster predictions.

Regarding compute, this experiment can be generated in a few minutes using a single GTX 2080 Ti.

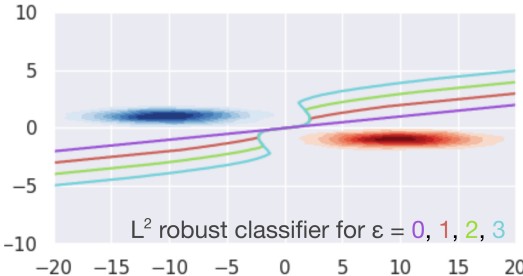

Figure 6: Decision boundary of our model at multiple levels of robustness on an example from Ilyas et al. [29].

# I Experimental details of adversarial experiments

Results for CIFAR-10 and ImageNet experiments comparing to state-of-the-art methods can be found in tables 7, 8, and 9. Table 7 only includes results for Randomized Smoothing (RS).

**Hyper-parameters and other details of the experiments** there are just three hyper-parameters to tweak for these experiments, as we try to remain as close as possible to the experiments from Cohen et al. [14]. In particular, we tried different added noises $\nu \in [0.05, 0.1, 0.2]$ and tailoring inner steps $\lambda \in [10^{-3}, 10^{-2}, 10^{-1}, 10^0]$ for $\sigma = 0.5$. To minimize compute, we tried these settings by tailoring (not meta-tailoring) the original model and seeing its effects on the smoothness and stability of optimization, choosing $\nu = 0.1, \lambda = 0.1$ (the fact that they're the same is a coincidence). We chose to only do a single tailoring step to reduce the computational burden, since robustness certification is very expensive, as each example requires 100k evaluations (see below). For simplicity and to avoid

| $\sigma$ | Method | 0.0 | 0.25 | 0.5 | 0.75 | 1.0 | 1.25 | 1.5 | 1.75 | 2.00 | 2.25 | ACR |
|---|---|---|---|---|---|---|---|---|---|---|---|---|
| 0.25 | (Inductive) RS | 0.75 | 0.60 | 0.43 | 0.26 | 0.00 | 0.00 | 0.00 | 0.00 | 0.00 | 0.00 | 0.416 |
| | Meta-tailor RS | **0.80** | **0.66** | **0.48** | 0.29 | 0.00 | 0.00 | 0.00 | 0.00 | 0.00 | 0.00 | **0.452** |
| 0.50 | (Inductive) RS | 0.65 | 0.54 | 0.41 | 0.32 | 0.23 | 0.15 | 0.09 | 0.04 | 0.00 | 0.00 | 0.491 |
| | Meta-tailor RS | 0.68 | 0.57 | 0.45 | **0.33** | 0.23 | 0.15 | 0.08 | 0.04 | 0.00 | 0.00 | **0.542** |
| 1.00 | (Inductive) RS | 0.47 | 0.39 | 0.34 | 0.28 | 0.21 | 0.17 | **0.14** | 0.08 | 0.05 | 0.03 | 0.458 |
| | Meta-tailor RS | 0.50 | 0.43 | 0.36 | 0.30 | **0.24** | **0.19** | **0.14** | **0.10** | **0.07** | **0.05** | **0.546** |

Figure 7: Percentage of points with certificate above different radii, and average certified radius (ACR) for on the CIFAR-10 dataset. Meta-tailoring improves the Average Certification Radius by $8.6\%, 10.4\%, 19.2\%$ respectively. Results for Cohen et al. [14] are taken from Zhai et al. [53] because they add more measures than the original work, with similar results.

| $\sigma$ | Method | 0.0 | 0.25 | 0.5 | 0.75 | 1.0 | 1.25 | 1.5 | 1.75 | 2.00 | 2.25 | ACR |
|---|---|---|---|---|---|---|---|---|---|---|---|---|
| 0.25 | RandSmooth | 0.75 | 0.60 | 0.43 | 0.26 | 0.00 | 0.00 | 0.00 | 0.00 | 0.00 | 0.00 | 0.416 |
| | Salman | 0.74 | 0.67 | 0.57 | 0.47 | 0.00 | 0.00 | 0.00 | 0.00 | 0.00 | 0.00 | 0.538 |
| | MACER | **0.81** | **0.71** | **0.59** | 0.43 | 0.00 | 0.00 | 0.00 | 0.00 | 0.00 | 0.00 | **0.556** |
| | Meta-tailored | 0.80 | 0.66 | 0.48 | 0.29 | 0.00 | 0.00 | 0.00 | 0.00 | 0.00 | 0.00 | 0.452 |
| 0.50 | RandSmooth | 0.65 | 0.54 | 0.41 | 0.32 | 0.23 | 0.15 | 0.09 | 0.04 | 0.00 | 0.00 | 0.491 |
| | Salman | 0.50 | 0.46 | 0.44 | 0.40 | 0.38 | 0.33 | 0.29 | 0.23 | 0.00 | 0.00 | 0.709 |
| | MACER | 0.66 | 0.60 | 0.53 | **0.46** | 0.38 | 0.29 | 0.19 | 0.12 | 0.00 | 0.00 | **0.726** |
| | Meta-tailored | 0.68 | 0.57 | 0.45 | 0.33 | 0.23 | 0.15 | 0.08 | 0.04 | 0.00 | 0.00 | 0.542 |
| 1.00 | RandSmooth | 0.47 | 0.39 | 0.34 | 0.28 | 0.21 | 0.17 | 0.14 | 0.08 | 0.05 | 0.03 | 0.458 |
| | Salman | 0.45 | 0.41 | 0.38 | 0.35 | **0.32** | 0.28 | **0.25** | **0.22** | **0.19** | **0.17** | 0.787 |
| | MACER | 0.45 | 0.41 | 0.38 | 0.35 | **0.32** | 0.29 | 0.25 | 0.22 | 0.18 | 0.16 | **0.792** |
| | Meta-tailored | 0.50 | 0.43 | 0.36 | 0.30 | 0.24 | 0.19 | 0.14 | 0.10 | 0.07 | 0.05 | 0.546 |

Figure 8: Percentage of points with certificate above different radii, and average certified radius (ACR) for on the CIFAR-10 dataset, comparing with SOA methods. In contrast to pretty competitive results in ImageNet, meta-tailoring improves randomized smoothing, but not enough to reach SOA. It is worth noting that the SOA algorithms could also likely be improved via meta-tailoring.

| $\sigma$ | Method | 0.0 | 0.5 | 1.0 | 1.5 | 2.0 | 2.5 | 3.0 | ACR |
|---|---|---|---|---|---|---|---|---|---|
| 0.25 | RandSmooth | 0.67 | 0.49 | 0.00 | 0.00 | 0.00 | 0.00 | 0.00 | 0.470 |
| | Salman | 0.65 | 0.56 | 0.00 | 0.00 | 0.00 | 0.00 | 0.00 | 0.528 |
| | MACER | 0.68 | **0.57** | 0.00 | 0.00 | 0.00 | 0.00 | 0.00 | **0.544** |
| | Meta-tailored RS | **0.72** | 0.55 | 0.00 | 0.00 | 0.00 | 0.00 | 0.00 | 0.494 |
| 0.50 | RandSmooth | 0.57 | 0.46 | 0.37 | 0.29 | 0.00 | 0.00 | 0.00 | 0.720 |
| | Salman | 0.54 | 0.49 | **0.43** | **0.37** | 0.00 | 0.00 | 0.00 | 0.815 |
| | MACER | 0.64 | 0.53 | **0.43** | 0.31 | 0.00 | 0.00 | 0.00 | **0.831** |
| | Meta-tailored RS | 0.66 | 0.54 | 0.42 | 0.31 | 0.00 | 0.00 | 0.00 | 0.819 |
| 1.00 | RandSmooth | 0.44 | 0.38 | 0.33 | 0.26 | 0.19 | 0.15 | 0.12 | 0.863 |
| | Salman | 0.40 | 0.38 | 0.33 | 0.30 | **0.27** | **0.25** | **0.20** | 1.003 |
| | MACER | 0.48 | 0.37 | 0.34 | 0.30 | 0.25 | 0.18 | 0.14 | 1.008 |
| | Meta-tailored RS | 0.52 | 0.45 | 0.36 | 0.31 | 0.24 | 0.20 | 0.15 | **1.032** |

Figure 9: Percentage of points with certificate above different radii, and average certified radius (ACR) for on the ImageNet dataset, including other SOA methods. Randomized smoothing with meta-tailoring are very competitive with other SOA methods, including having the biggest ACR for $\sigma = 1$.

excessive tuning, we chose the hyper-parameters for $\sigma = 0.5$ and copied them for $\sigma = 0.25$ and $\sigma = 1$. As mentioned in the main text, $\sigma = 1$ required initializing our model with that of Cohen et al. [14] (training wasn't stable otherwise), which is easy to do using CNGRAD.

In terms of implementation, we use the codebase of Cohen et al. [14](`https://github.com/locuslab/smoothing`) extensively, modifying it only in a few places, most notably in the architecture to include tailoring in its forward method. It is also worth noting that we had to deactivate their disabling of gradients during certification, because tailoring requires gradients. We chose to use the first-order version of CNGRAD which made it much easier to keep our implementation very close to the original. It is likely that doing more tailoring steps would result in better performance.

We note that other works focused on adversarial examples, such as Zhai et al. [53], Salman et al. [43], improve on Cohen et al. [14] by bigger margins. However, tailoring and meta-tailoring can also improve a broad range of algorithms in applications outside of adversarial examples. Moreover, they could also improve these new algorithms further, as these algorithms can also be tailored and meta-tailored.

**Compute requirements** For the CIFAR-10 experiments building on Cohen et al. [14], each training of the meta-tailored method was done in a single GTX 2080 Ti for 6 hours. Certification was much more expensive (10k examples with 100k predictions each for a total of $10^9$ predictions). Since certifications of different images can be done in parallel, we used a cluster consisting of 8 GTX 2080 Ti, 16 Tesla V-100, and 40 K80s (which are about 5 times slower), during 36 hours.

For the ImageNet experiments, we fine-tuned the original models for 5 epochs; each took 18 hours on 1 Tesla V-100. We then used 30 Tesla V-100 for 20 hours for certification.