# OpenReview forum: "Tailoring: encoding inductive biases by optimizing unsupervised objectives at prediction time"
_NeurIPS.cc/2021/Conference — NeurIPS 2021 Poster_

### Official Review · Reviewer_3ay3 · 2021-07-04

**Rating:** 7
**Confidence:** 4

**Summary:**

In this paper, the authors propose optimizing an unsupervised loss function at test time as an inductive prior on a neural network. They propose two schemes for achieving this: training a network in a regular fashion and then applying the unsupervised loss at inference time only, or applying the unsupervised loss during training as a meta-learning scheme. The authors claim that this method can improve robustness, as well as improve train-test generalization gap in cases where there is a known inductive prior on the expected output (e.g. when modelling physical systems)


**Limitations And Societal Impact:**

The authors have adequately addressed the limitations of their work and Societal impact.

**Main Review:**

The paper's writing could be improved, as the writing is complex and hard to follow. Furthermore the model details were mixed with the introduction, making it hard to understand what is the author's proposed work. It is also hard to understand the implementation details of this idea, as the authors share neither pseudo-code nor actual code to illustrate their implementation. With such an idea, it is often the case that the devil is in the details, and efforts to replicate it could show very different results to the ones shown on the paper.

The idea is very interesting and novel. It is well motivated in learning theory and seems almost obvious in hindsight (a good thing!). The authors provide a wide variety of experiments to show the effectiveness of the idea, not just for different datasets, but for different types of inductive bias. Especially in the case of physical systems modelling, this idea could be very useful in the applied setting and have a broad impact on further research.

**Time Spent Reviewing:**

1

---

> ### Author Response · Authors · 2021-08-11
> **Thank you for your review**
>
> Thank you for your review.
>
>
> **Complexity of the writing** We will improve the clarity of the text. In particular, after seeing the feedback from all reviewers we will devote more space to the method and experiment sections. Since space is limited, we plan on moving to the appendix some details of the comparison between meta-tailoring and meta-learning as well as the broader impact statement.
>
>
> **Code and pseudo-code** Pseudo-code can be found in appendix D, pages 22 & 23. We will open-source the code for the paper to facilitate reproducibility.

---

### Official Review · Reviewer_7twm · 2021-07-06

**Rating:** 5
**Confidence:** 3

**Summary:**

The paper proposes a new method to adapt neural networks at inference time to the given input such that the model minimizes some given unsupervised loss. This method is called 'tailoring'. Additionally, 'meta-tailoring' is proposed that trains the model using 'tailoring', too. This way the gap between training and inference distribution is removed that 'tailoring' introduces.

**Limitations And Societal Impact:**

The authors consider interesting societal impacts and the limitations of their algorithm.

**Main Review:**

The paper describes a, to the best of my knowledge, novel method. I believe this method is interesting for niche applications of ML. The model is replaced by a short SGD loop of the model, such that given x a few iterations of SGD are performed, before making the prediction. I do not understand why there is no possibility to make the model behave straight away like the model after a few steps.

Main Points:

1. Method Description. The method is not described enough in the main paper. One has to read the appendix to really understand what the method is, especially for meta-tailoring. There should be pseudo-code or at least a more detailed text description in the main paper. For example, I do not understand what the difference of Meta-Tailoring (0 st.) is to the baseline, even after looking into the appendix. I am pretty confident that algorithm 1 would actually not train at all with steps=0. I also find the naming confusing with the introduction of CNGrad, but naming the application of CNGrad in the experiments Meta-tailoring again.

2. First-order and detached CNGrad. You only consider detached CNGrad in all experiments (line. This is not the algorithm you provide guarantees for in Section 3. I think this is a severe short-coming, as the detached variant, also does not agree with the intuition one develops around meta-tailoring. For 2/4 experiments you write that you use first-order CNGrad, for the others it is not known. First-order CNGrad, goes even further away from meta-tailoring and detaches \gamma and \beta even earlier. This, thus makes the above problem even more severe.

3. Focus. I think this paper could benefit a lot of a more focused structure. You try to keep everything as general as possible, but there you also have to make changes for each experiment. I believe the paper might benefit from more focus. Only considering one or two CNGrad variants with fixed hyper-parameters across tasks, a main evaluation on which you reach something close to state-of-the-art and more focus on the method presentation.

4. Experiments. This paper proposes a broad method, thus I understand the motivation of the authors to include a diverse set of benchmarks. The problem with this setup is that, the evaluations themselves suffer from this. None, of the evaluations is described to a sufficient level, such that it is really hard to understand the effect.

5. Baselines. The baselines are not easily understood. While, I very much appreciate that you add the tailoring loss to the inductive baseline for the first experiment, I am still a little critical regarding the baselines. You now put a lot of work in making CNGrad work including selecting where to apply CN layers, what inner-lr to use, whether to use first-order CNGrad or detached CNGrad, but for the baselines it seems you performed less of this tuning. For some of the baselines my feeling is that it is reasonably easy to come up with a way to make the inductive baseline perform well and consider the constraints at testing time. Like you wrote for Adverserial Examples.

6. Costs. As this method introduces a considerable amount of extra gradient steps to the training loop, I believe there should be some considerations of the costs of this method in the main paper.

Summary: The paper proposes an interesting new method. It does not show conclusive evidence that it improves state-of-the-art models in any domain, though, and it uses a different algorithm compared to the described algorithm. Additionally, parts of the paper seem to not be quite ready for a main conference and rather in a draft-stage. Nevertheless, I believe this paper might be a great contribution after some more work on the experiments and the presentation.

DISCLAIMER: I did not take the time to read section 3 and the corresponding proofs in detail, as I do not believe that this changes my view of this paper too much, it is hard to follow and does, to my understanding, not apply to the algorithm actually used practically in the experiments as pointed out in 2.

Details:

i) Line 720 'key' -> 'this is key'

ii) Line 105 '. Losses' -> '.\n\nLosses'

iii) Line 63 -> Notice that the outer process now only optimizes the objective we care about, ...

iv) Confusing notation in algorithms, with var assignments and 'for' construct in the same line.

v) Table 1 description 'over-performs' -> 'outperforms'

vi) You used /begin{figure} for many tables where /begin{table} should be used.

vii) Line 356 ' Improving' -> 'Improving'

**Time Spent Reviewing:**

3

---

> ### Author Response · Authors · 2021-08-11
> **Thank you for your review**
>
> Thank you for the detailed review and constructive comments.
>
> **Clarifications on 0-step meta-tailoring** We agree that the notation was confusing; we will clarify it. The number of steps within the parentheses of table 1 corresponds only to the number of inner steps taken at test time. For meta-tailoring, we performed 2 inner steps during training and then analyzed the performance of varying the test-time inner steps. This analysis is common for gradient-based meta-learning algorithms, where we meta-train with a fixed number of inner steps and observe how performance evolves w.r.t. the number of meta-test inner steps. In table 1, 0-step meta-tailoring refers to the case where we perform 0 inner steps at test time, but have performed 2 inner steps at training time. We will clarify this in the main text.
>
> **Devoting more space to algorithm, experiments and compute costs** We agree that more detail in the method section and the experiments would be fruitful. Given the amount of theoretical content and number of experiments, we had to move many details to the appendix, as well as an entire toy-but-insightful experiment (appendix H). We are considering moving other subsections to the appendix to increase the details in the method and experiments sections. In particular, we will summarize the relationship between meta-tailoring and meta-learning on page 3 and the broader impact statement. We welcome other suggestions!
> With respect to computational costs, note that we discuss them in section 6.2 in the main text, with extra details in lines 725-738 in appendix D. We will move some of these details to the main text.
>
> **CNGrad vs meta-tailoring** Meta-tailoring computes a different set of weights for each example. In PyTorch and TensorFlow, we cannot parallelize this process in the general case (it is possible to do it in JAX). CNGrad allows us to parallelize meta-tailoring in all these platforms, by only customizing the conditional normalization layers to each example. Tailoring and meta-tailoring are the frameworks, CNGrad is an architectural trick to make them efficient. This is why we simultaneously refer to meta-tailoring and CNGrad in our experiments. We will make it clearer in the text.
>
> **First-order CNGrad** The reviewer correctly points out that all results come from first-order CNGrad (which is the same as detached CNGrad). However, note that we also tried CNGrad’s  second-order version for the first experiment, which actually performed slightly worse. This was not entirely surprising for the following reasons:
> As observed in the meta-learning literature, second-order gradients are often unstable, requiring us to use a smaller inner learning rate.
> As in the case of FO-MAML[1] or REPTILE[2] for meta-learning, first-order CNGrad does not warp the inner optimization but still takes the adaptation into account. Therefore it still satisfies the essence of meta-tailoring captured in the theory section(#3).
> First-order versions have proven successful in meta-learning[1,2]. In meta-tailoring, all networks tackle the same task. Therefore, we expect the differences between the tailored weights of each example to be smaller than the differences between the adapted weights of different tasks in meta-learning. Since the second-order part of the optimization concerns itself with these small differences between examples, the importance of second-order gradients in meta-tailoring will be even smaller than in meta-learning.
> Given their similar performance, we used the first-order version of CNGrad because it is easier to implement and faster. An in-depth comparison between both options can be found in lines 693-710 in appendix D.
>
> **Varying hyper-parameters and strength of the baselines** The reviewer rightfully points out that meta-tailoring with CNGrad adds hyper-parameters: number of inner steps, inner learning rate, and whether to use first-order or second-order optimization in CNGrad. However, this doesn’t result in stronger hyperparameter tuning w.r.t the baselines for the following reasons:
> - **Good baseline tuning**: all but four baselines come from impactful published works, which were optimized by their corresponding authors. We took great care with the remaining four baselines implemented by us. We thank the reviewer for noticing that we appropriately tuned the corresponding hyperparameter for the two inductive baselines in the physics experiment. In that experiment, we also allowed the TTT baseline to do the inner loop up to convergence. This used 5 times more steps than meta-tailoring (the more steps the better for both methods, as seen in table 1 and figure 2). Finally, for the TTT baseline for the contrastive learning experiments we searched over steps and learning rate in the exact same way as for meta-tailoring (this tuning is shown in figure 5 in appendix G).
> - **Monotonic dependence w.r.t. hyperparameters**: as long as the optimization is stable (which can be easily checked during training) we observe a mostly monotonic relationship between the final task loss and the three hyper-parameters. A higher learning rate is better, more steps are better, and second-order is better than first-order. Increasing the inner learning rate eventually resulted in instabilities, both the steps and second-order CNGrad increase compute, and the second-order CNGrad increased code complexity. Since we were not seeking the state-of-the-art, we always used the first-order version and a small number of steps to reduce compute and code complexity. Using more steps or the second-order version could have improved results further.
> - **Diversity of experiments results in varied hyperparameters**: all experiments use first-order CNGrad. The reviewer is correct in mentioning that the number of steps and inner learning rate vary across experiments. This is because this paper includes a wide diversity of applications. Some networks are very small (3-layers) and some are large (ResNet-50 in the contrastive and adversarial experiments). Having a small network allows having a larger number of steps. Similarly, losses also vary greatly: some are MSE and some are cross-entropy, with varied scales. These different magnitudes affect the range of effective inner learning rates.
>
>
> [1] Model-Agnostic Meta-Learning for Fast Adaptation of Deep Networks; Finn et al. ‘17
>
> [2] On First-Order Meta-Learning Algorithms; Nichol et al. ‘18

---

> > ### Comment · Reviewer_7twm · 2021-08-16
> > **Updated Opinions**
> >
> > On 0-step meta-tailoring: Oh, I understand now. That is interesting. Updating the BatchNorm parameters during training for each example yields a neural network that generalizes better even without these updates.
> >
> > CNGrad vs meta-tailoring: It seems to only be a communication problem, e.g. in the contrastive learning experiments you don't do CNGrad, but aren't explicit about it. But I am confused again: in algorithm 1 you actually take steps with respect to $w$ $steps$ times. Shouldn't you only take one final step with $w$, and $steps$ many steps with $\beta$ and $\gamma$ only?
> >
> > The Math: I believe the math of the method you do not even use has too big a part of this paper. If you had the extra page to explain your method and the experiments in detail the paper would be stronger in my opinion.
> >
> > The hyper-parameters: The monotonicity, of course, is beautiful and makes my arguments less strong.
> >
> > Overall, I am scared this paper does not provide a contribution that can be built on. This is mostly due to the communication and could therefore likely be fixed without further experiments.

---

> > > ### Author Response · Authors · 2021-08-17
> > > **Thank you for your reply**
> > >
> > > Thank you for your fast reply.
> > >
> > > **0-step meta-tailoring**: Yes, exactly!
> > >
> > > **CNGrad vs meta-tailoring**: All experiments, including contrastive learning, apply meta-tailoring using first-order CNGrad (which allows it to be run efficiently). It’s similar to saying a paper does meta-learning using FO-MAML[1]. In the same way, CNGrad is a meta-tailoring algorithm.
> > >
> > > Your observation about algorithm 1 is a great and subtle one; both options are correct! Taking a single outer step after all inner steps is similar to MAML[1].  Taking an outer step after every inner step is similar to WarpGrad[2], which got better results than MAML by having some weights $w$ trained in the outer loop and some weights $w’$ trained in the inner loop. CNGrad builds on WarpGrad with the customization of the weights $w’$ being CN layers, which allows the efficient parallelization. Preliminary experiments didn’t show much difference between both approaches (MAML-like vs. WarpGrad-like). Therefore, we chose the latter for consistency.
> > >
> > > **Math applies to all the experiments**: we want to emphasize that section 3 does apply to all the experiments. As described in theorem 1, it applies to any meta-tailoring algorithm (where the prediction function is the same for test and train). Theorem 1 and Remark 1 provide upper-bounds of the test task loss that depend on $\mathcal{L}^{tailor}(x,\theta_{x,S})$. The same bounds apply to regular inductive learning algorithms by just changing $\theta_{x,S}$ to $\theta_S$. The key insight then is that meta-tailoring algorithms can optimize $\mathcal{L}^{tailor}(x,\theta_{x,S})$ for each $x$ at prediction time, lowering the upper-bound.  Because all experiments use meta-tailoring, they are all within the scope of the theory results.
> > >
> > > **Clarity** Thank you for all your questions and comments. We believe they will greatly improve the clarity of the manuscript. For instance, we will move some parts of the theory to the appendix and bring information on the algorithm from appendix D to the main text.
> > >
> > > [1] Model-agnostic meta-learning for fast adaptation of deep networks; Finn et al. ‘17
> > >
> > > [2] Meta-Learning with Warped Gradient Descent; Flennerhag et al. ‘19

---

> > > > ### Author Response · Authors · 2021-08-18
> > > > **Update: pseudo-code added to the main text**
> > > >
> > > > Following your suggestions, we've now changed theory content for method content.
> > > >
> > > > To make space, we have moved some math(Assumption 1, Theorem 2, and details from section 3) to the appendix. These were mostly math technicalities and not part of the main punchline. Therefore, we think the clarity of the theory has not decreased.
> > > >
> > > > This allowed us to move the entire Algorithm pseudo-code (both training and prediction) from App D to the main text along with some extra comments on the method. We think this will greatly benefit clarity as it will make the method much more concrete to the reader.
> > > >
> > > > Thanks for the suggestion!

---

### Official Review · Reviewer_66Ub · 2021-07-11

**Rating:** 7
**Confidence:** 3

**Summary:**

The paper proposes tailoring - a general framework of algorithms that can combine ideas for test-time generalization, self supervision, meta learning and transductive learning. Although the experiments are quite limited overall, the paper is a good addition for the ML community.

**Limitations And Societal Impact:**

The most practical problems with the frameworks (as the authors discuss) is with privacy as the models are "tailored" for the given example. This can have negative impact and which the authors correctly discuss.

**Main Review:**

The paper introduces tailoring and meta tailoring as a means to add inductive biases at test-time using contrastive losses. The paper overcomes some of the shortcomings of TTT and meta-tailoring seems like an interesting improvement over TTT. The idea of encouraging soft inductive biases (5.2) is very interesting and practical. I would like the authors to include experiments such as domain generalization (like the sort done by TTT) since I believe that those set of experiments are very good tests of how such algorithms can adapt to novel data distributions (not just adversarial samples as they are a very specific type of generalization. Although the authors mention the practical problems with implementation in popular deep learning frameworks (pytorch and tensorflow), it would be good for the authors to provide means to overcome such problems so that CNGrad can become a staple in the deployment of ML models.

The authors tend to focus on the breadth of results to show the generality of the solution, rather than depth in one or two fields, it might be useful to show more difficult tasks in any of the tasks to show a strict improvement and the scalability of the solution.

**Time Spent Reviewing:**

1 hour

---

> ### Author Response · Authors · 2021-08-11
> **Thank you for your review**
>
> Thank you for your review.
>
> **Breadth of applicability** We decided to do many experiments to display the breadth of applicability of tailoring as a way to encode a wide variety of inductive biases in the standard ML setting. It is worth noting that we chose to build on some of the most impactful papers in contrastive learning, certified adversarial examples, and physics modeling to facilitate understanding instead of using the state-of-the-art works, which often have more bells and whistles. Moreover, all three are still close to SOA in their respective fields. We also chose to focus on the same-distribution setting, where our theoretical guarantees apply, to keep the message clear and consistent.
>
> **Practical implementation in deep learning frameworks** Both tailoring and meta-tailoring are already efficiently implementable in JAX. This is because JAX allows the evaluation of multiple networks with the same architecture but different weights in a batch. To the best of our knowledge, this is not yet possible in PyTorch and TensorFlow. For these frameworks, we constructed CNGrad, where parameter updates only occur in conditional normalization layers, which can be efficiently parallelized. Therefore, CNGrad can be efficiently deployed in all popular deep learning frameworks. We will clarify this in the text.

---

### Official Review · Reviewer_Pr63 · 2021-08-03

**Rating:** 6
**Confidence:** 4

**Summary:**

This paper proposed tailoring, which is a general framework that could help to finetune the prediction on each test sample according to some specific inductive biases. Tailoring provides a different perspective that avoids involving extra loss function in the proxy fashion.  Besides, the authors proposed meta-tailoring, which integrates the unsupervised loss in a way similar to meta-learning. The theoretical discussion and empirical results demonstrate the effectiveness of the proposed tailoring.

**Limitations And Societal Impact:**

Yes.
Refers to main review

**Main Review:**

Strengths:
1. The idea of tailoring is interesting. The current neural networks mainly conduct amortized optimization, how to reduce such gap especially on the test data is an important research direction. I believe the proposed framework could provide a different perspective on understanding the "generalization" gap and different inductive biases.

2. The limitation of tailoring lies in the increased computational cost. To reduce such extra cost, the authors thus introduce CNGRAD which could efficiently parallel the evaluation of the model over multiple samples.  And there is detailed and sound theoretical justification on the CNGRAD provided.

3. The authors provided extensive examples of inductive biases. And correspondingly the experiment results on symmetry constraints, inductive biases, contrastive loss, and adversarial examples justify the effectiveness of both tailoring and meta-tailoring.  The ablation study is well designed and the results are promising.

Weakness:
1. One particular concern of mine is when conducting tailoring the parameters of the model change according to a single sample, this is similar to the continual learning setting. I wonder how the methods could avoid catastrophic forgetting. It seems that the authors constrain the steps of the tailoring while small changes in parameter space could result in the relatively large change of the model output. I suggest the authors add more discussions on this part.

2. Though the method does no limit the application scenarios, I feel that the method is highly related to the field of semi-supervised learning. Therefore, I suggest more discussions on the related works. For example, the adversarial examples setting of tailoring is related the virtual adversarial learning in semi-supervised learning [1].

[1]. Virtual Adversarial Training: A Regularization Method for Supervised and Semi-Supervised Learning

Questions:
I am curious about the setting when evaluating the test sample, we first assign a pseudo label according to the initial output. And minimizing the loss towards this pseudo label, I wonder whether tailoring in this setting could work.

**Time Spent Reviewing:**

4 hours

---

> ### Author Response · Authors · 2021-08-11
> **Thank you for your review**
>
> **Avoiding catastrophic forgetting** Partial catastrophic forgetting of the task loss can occur when doing tailoring with a large learning rate or for many inner steps. However, it does not occur when doing meta-tailoring. This can be seen in figure 2. The green curve (representing tailoring) begins at the top right and, at first, goes down and to the left: as we minimize the physics tailoring loss, the task loss also decreases. However, as you mention, the task loss eventually starts to increase after many inner steps. Therefore, tailoring is mostly useful for a few inner steps (which will also be faster). In the case of meta-tailoring, catastrophic forgetting is not a problem because meta-tailoring takes the tailoring process into account during training. There, the model is trained to have a low task loss after the tailoring update and thus learns not to forget about the task loss after the inner optimization. You can see that in the monotonicity of the blue curve.
>
> **Relation to semi-supervised learning** We agree about the relevance of the semi-supervised learning literature and virtual adversarial training. We will add more discussion on both topics. It is worth noting that transductive learning (which we list as one of our main inspirations) is closely related to semi-supervised learning, as it assumes unlabeled test data is available at training time. We will clarify this relationship in the paper; following the great analysis of chapters 24&25 in [1].
>
> **Tailoring with pseudo-labels** Allowing the model to make an initial prediction, then tailoring the model to maximize its confidence makes a lot of sense. In particular, minimizing entropy at the test points was the main loss used in the classic transductive learning literature. Your idea would be to bring this loss to the meta-tailoring setting, which is likely to be useful!
>
> [1] Semi-Supervised Learning; Chapelle, Scholkopf, Zien

---

> > ### Comment · Reviewer_Pr63 · 2021-08-31
> > **Response to Rebuttal**
> >
> > Thanks for your responses.
> > >Avoiding catastrophic forgetting Partial catastrophic forgetting ........ There, the model is trained to have a low task loss after the tailoring update and thus learns not to forget about the task loss after the inner optimization. You can see that in the monotonicity of the blue curve.
> >
> > I see.  Selecting good tailoring loss still seems to be a little tricky, or it could be contrastive to the task loss. Is it possible to provide more guidance on how to select tailoring loss?
> >
> > The rebuttal fixed most of my concerns, I prefer to keep my score and vote for acceptance.

---

> > > ### Author Response · Authors · 2021-08-31
> > > **Selecting a good tailoring loss for tailoring and why meta-tailoring is robust to this choice**
> > >
> > > That's a good question! In this paper, we suggest tailoring losses serve a similar role to auxiliary losses, but better serve the outer objective and, in the experiments shown in the paper, work better. Intuitively, you want tailoring losses to be informative of the true task, so that the gradient minimizing the tailoring loss is aligned with the gradient minimizing the task loss (because at test time you can only see the former, but not the latter). This was analyzed for auxiliary losses [which are widely used] in [1].
> > >
> > > Finally, one notable point is that in meta-tailoring the network will be trained to perform well _after_ the update. Therefore, in the same way Figure 2 shows it doesn't overfit to the tailoring loss, if the tailoring loss is detrimental meta-tailoring is likely to ignore it.
> > >
> > > We will add a comment (and pointer to [1]) in the main text.
> > >
> > > [1] Adapting auxiliary losses using gradient similarity; Du*,Czarnecki* et al.  '2020; https://arxiv.org/abs/1812.02224

---

### Decision · Program_Chairs · 2021-09-27

**Decision:**

Accept (Poster)

**Comment:**

This paper proposes tailoring, a technique for incorporating unsupervised objectives during test time. The main distinction and past work is that tailoring is applied to each individual test datapoint. The paper includes some theoretical justification of the approach as well as rather extensive experiments. Reviewers were generally positive on the paper. The main criticism of the paper came down to lack of clarity in writing, but the authors have addressed this concern through their rebuttal. Before the camera-ready version, please incorporate the proposed changes to writing.